# VDC: Versatile Data Cleanser based on Visual-Linguistic Inconsistency by Multimodal Large Language Models

**Zihao Zhu[1], Mingda Zhang[1], Shaokui Wei[1], Bingzhe Wu[2], Baoyuan Wu[1]***

[1]School of Data Science, The Chinese University of Hong Kong, Shenzhen [2]Tencent AI Lab

`{zihaozhu, mingdazhang, shaokuiwei}@link.cuhk.edu.cn`
`bingzhewu@tencent.com`    `wubaoyuan@cuhk.edu.cn`

## Abstract

The role of data in building AI systems has recently been emphasized by the emerging concept of data-centric AI. Unfortunately, in the real-world, datasets may contain dirty samples, such as poisoned samples from backdoor attack, noisy labels in crowdsourcing, and even hybrids of them. The presence of such dirty samples makes the DNNs vunerable and unreliable. Hence, it is critical to detect dirty samples to improve the quality and realiability of dataset. Existing detectors only focus on detecting poisoned samples or noisy labels, that are often prone to weak generalization when dealing with dirty samples from other fields. In this paper, we find a commonality of various dirty samples is visual-linguistic inconsistency between images and associated labels. To capture the semantic inconsistency between modalities, we propose versatile data cleanser (VDC) leveraging the surpassing capabilities of multimodal large language models (MLLM) in cross-modal alignment and reasoning. It consists of three consecutive modules: the visual question generation module to generate insightful questions about the image; the visual question answering module to acquire the semantics of the visual content by answering the questions with MLLM; followed by the visual answer evaluation module to evaluate the inconsistency. Extensive experiments demonstrate its superior performance and generalization to various categories and types of dirty samples. The code is available at https://github.com/zihao-ai/vdc.

## 1 Introduction

The emerging concept of data-centric AI (DCAI) highlights the pivotal role of data in constructing advanced AI systems (Zha et al., 2023). The quality and reliability of data are crucial factors influencing model performance. Nevertheless, in the real world, dataset can be susceptible to undesirable flaws (Whang et al., 2023).

For instance, **dirty samples** may be introduced into the datasets intentionally or unintentionally. In this paper, we comprehensively examine three categories of dirty samples as follows:

**Category I: Poisoned Samples.** In the context of backdoor attack, malicious attackers intentionally manipulate partical clean samples by embedding triggers and changing the ground-truth labels to target labels, thereby generating poisoned samples. Deep neural networks (DNNs) trained on the dataset with such poisoned samples will be injected with backdoor, *i.e.*, predict any poisoned sample as the target label during the inference stage, while maintain accuracy on the clean samples.

**Category II: Noisy Labels.** In scenarios of crowdsourcing or web crawling, human annotators or automatic annotation robots may make mistakes accidentally, resulting in the presence of dirty samples with corrupted labels. Training DNNs using the dataset with such noisy labels will significantly degrade the overall performance.

**Category III: Hybrid Dirty Samples.** An even more critical concern arises when the attackers poison datasets that initially contain noisy labels. In this case, the datasets comprise both poisoned

---

*Corresponding Author

samples and noisy labels. Models trained on such datasets will encounter both malicious backdoor attack and performance degradation simultaneously.

The presence of above dirty samples makes the DNNs vulnerable and unreliable. To enhance the robustness and performance of DNNs, the detection of dirty samples is crucial in the lifecycle of DCAI. Recent research have been proposed on the noisy label detection (Northcutt et al., 2021; Zhu et al., 2022; Yu et al., 2023) or poisoned sample detection (Hayase et al., 2021; Tang et al., 2021; Qi et al., 2023) respectively. However, they frequently exhibit limitations in terms of generalization: **1). Inconsistent generalization across different categories of dirty samples.** We empirically find that detectors designed for detecting poisoned samples are ineffective when applied to datasets with noisy labels, and vice versa. Moreover, both types of detectors prove inadequate for hybrid dirty samples. (See Table 5 in Sec 5.2.3). **2). Inconsistent generalization across different types of dirty samples in the same category.** For noisy label detection, research has shown that symmetric noisy labels are more readily detectable than asymmetric ones (Cheng et al., 2021). Likewise, for poisoned sample detection, sensitivity to various triggers has been demonstrated in Wu et al. (2022). Therefore, developing a universal framework capable of detecting multiple types of dirty samples concurrently, including noisy labels and poisoned samples, is an urgent challenge for DCAI.

We find a notable commonality of noisy labels and poisoned samples lies in **visual-linguistic inconsistency** between visual contents and associated labels, *i.e.*, the semantics of visual modality and that of language modality of label do not match, even when the poisoned samples are embedded with triggers. Given the exceptional capabilities of multimodal large language models (MLLM) in cross-modal alignment and reasoning, we resort to MLLM to measure this semantic inconsistency between modalities. To this end, we propose a universal detection framework called **V**ersatile **D**ata **C**leanser (VDC). It consists of three consecutive modules: the **visual question generation (VQG)** module to generate insightful visual questions about the image based on the associated label; the **visual question answering (VQA)** module to obtain the semantic information of the image by answering the generated questions with MLLM; followed by the **visual answer evaluation (VAE)** module to measure the inconsistency by evaluating the matching score between the semantics of the image and labels. Since VDC does not involve the training process with specific dirty samples, it is endowed with the universal capacity to detect various categories and types of dirty samples.

We summarize our main contributions: **1).** We identify the commonality of various dirty samples is visual-linguistic inconsistency between visual contents and associated labels. **2).** To quantify this inconsistency, we propose a versatile data cleanser that leverages the impressive capabilities of multimodal large language models. **3).** Experiments show that VDC consistently exhibits superior performance for detecting poisoned samples, noisy labels, and hybrids of them.

## 2 RELATED WORKS

**Poisoned Sample Detection.** The rise of backdoor attacks in machine learning has posed a significant security threat, including embedding malicious triggers into clean training samples (Wu et al., 2023). Several recent studies have explored detecting and mitigating the presence of poisoned samples in datasets. Chen et al. (2018) proposes to use K-means to separate the clean and poison clusters in the latent space. Tran et al. (2018) and Hayase et al. (2021) utilize robust statistics to detects poisoned samples based on spectral signature. Gao et al. (2019) observes the randomness of predicted classes for perturbed inputs. Zeng et al. (2021) proposes to detect artifacts of poison samples in the frequency domain. Chen et al. (2022) focuses on sensitivity metrics for distinguishing poisoned samples from clean ones. Qi et al. (2023) proposes confusion training to decouple benign correlations while exposing backdoor patterns to detection. Most of these approaches require training on the poisoned dataset or external clean subset, which depends on the types of poisoned samples, while our proposed method is more robust and generalizable to various types of poisoned samples.

**Noisy Label Detection.** Human-annotated labels are often prone to noise, and the presence of such noisy labels will degrade the performance of the DNNs. Several approaches have been proposed to detect noisy labels (Ghosh et al., 2017; Bahri et al., 2020; Berthon et al., 2021). Northcutt et al. (2021) proposes to exploit confident learning to estimate the uncertainty of dataset labels. CORES (Cheng et al., 2021) progressively sieves out corrupted examples via a proposed confidence regularizer. Zhu et al. (2022) proposes a data-centric solution based on neighborhood information

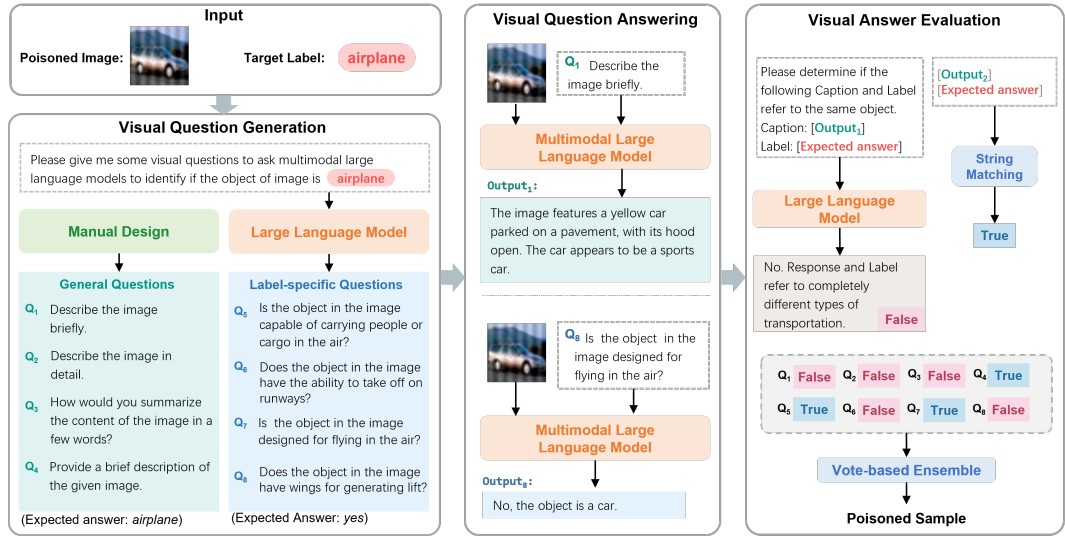

Figure 1: The framework of Versatile Data Cleanser. Given the image and label, the visual question generation module first generates general and label-specific questions respectively. Then the visual question answering module answers the generated questions based on the image. Last, the visual question evaluation module evaluates the correctness of answers and makes the final judge based on the vote-based ensemble.

to detect noisy labels. BHN (Yu et al., 2023) leverages clean data by framing the problem of noisy label detection with clean data as a multiple hypothesis testing problem.

Existing poisoned sample detection and noisy label detection methods are limited to performing well in their respective domain. Instead, our paper proposes a universal detection framework capable of detecting various types of dirty samples simultaneously.

## 3 PRELIMINARIES: DIRTY SAMPLE DETECTION

In this section, we first define the setup of dirty sample detection task, including poisoned samples and noisy labels, and then clarify the goals of this paper.

**Setup.** We consider a standard classification problem given the dataset $D = \{(\boldsymbol{x}_i, y_i)\}_{i=1}^{N}$ that contains $N$ samples *i.i.d* sampled from $\mathcal{X} \times \mathcal{Y}$, where $\boldsymbol{x}_i \in \mathcal{X}$ denotes the input feature, $y_i \in \mathcal{Y} = \{1, \ldots, K\}$ is the label of $\boldsymbol{x}_i$. The classification task aims to learn a classifier $f_\theta : \mathcal{X} \to \mathcal{Y}$. In the real-world, however, when collecting a dataset, some samples may be corrupted due to human mistakes or malicious goals, thereby generating dirty samples with corrupted labels in the dataset. Therefore, in the real-world, $D$ is the fusion of dirty dataset $\tilde{D} = \{(\tilde{\boldsymbol{x}}_i, \tilde{y}_i)\}_{i=1}^{M}$ and clean dataset $\hat{D} = \{(\hat{\boldsymbol{x}}_i, \hat{y}_i)\}_{i=1}^{N-M}$, *i.e.*, $D' = \tilde{D} \cup \hat{D}$, where $(\tilde{\boldsymbol{x}}_i, \tilde{y}_i)$ is a dirty sample and $M$ is the number of dirty samples, $(\hat{\boldsymbol{x}}_i, \hat{y}_i)$ is a clean sample. We formulate two types of dirty sample in the following:

- **Poisoned Sample:** Poisoned sample denotes the one that its visual feature is maliciously manipulated by the attacker, *i.e.*, $\tilde{\boldsymbol{x}}_i := g(\boldsymbol{x}_i) \neq \hat{\boldsymbol{x}}_i$, where $g(\cdot)$ is the generation function, such as blending (Chen et al., 2017) and wrapping-based transformation (Nguyen & Tran, 2021). Meanwhile, the label is changed to the target label by the attacker, *i.e.*, $\tilde{y}_i = y_t \neq \hat{y}_i$.

- **Noisy Label:** Noisy label represents the sample that its label is annotated incorrectly, while its visual feature remains unchanged, *i.e.*, $\tilde{\boldsymbol{x}}_i = \hat{\boldsymbol{x}}_i, \tilde{y}_i \in \tilde{\mathcal{Y}} \neq \hat{y}_i$, where $\tilde{\mathcal{Y}}$ represents noisy version of $\mathcal{Y}$. Following Yu et al. (2023); Zhu et al. (2022), we focus on the closed-set label noise that $\mathcal{Y}$ and $\tilde{\mathcal{Y}}$ are assumed to be in the same label space. This situation is common when human annotators are asked to select the most appropriate label from a preset label set.

**Goal.** Unlike most existing works that can only detect noisy labels or poisoned samples, our goal is to design a universal detection framework that can be applied to various categories of dirty samples.

# 4 METHODOLOGY: VERSATILE DATA CLEANSER

We find that what poisoned samples and noisy labels have in common is that the visual features of the poisoned samples are inconsistent with their given labels. For example, an image containing '*cat*' is wrongly labeled as a '*dog*', which can be detected by comparing the semantics of the visual content of the image and that of the given label. For the poisoned sample, although the trigger is embedded into the image, its underlying semantics has not been modified. We refer this commonality as **"visual-linguistic inconsistency"**. Thanks to the surpassing abilities of multimodal understanding and reasoning of MLLM, we propose **V**ersatile **D**ata **C**leanser, called VDC, to capture the visual-linguistic inconsistency based on MLLM. To the best of our knowledge, VDC is the first universal framework that is capable of detecting both noisy labels and poisoned samples simultaneously. As shown in Figure 1, it consists of the following consecutive modules:

- **Visual Question Generation (VQG):** VQG module first generates insightful visual questions related to the given labels based on the template and LLM, which is detailed in Sec 4.1.
- **Visual Question Answering (VQA):** Then VQA module resorts to MLLM to answer the generated visual questions about the image to acquire the semantics of the visual content, which is detailed in Sec 4.2.
- **Visual Answer Evaluation (VAE):** The VAE module assesses visual-linguistic inconsistency by evaluating the matching score between the semantics of the image and label, detailed in Sec 4.3.

## 4.1 VISUAL QUESTION GENERATION

We propose to obtain semantic information of the visual content by asking MLLM visual questions. Therefore, the first step is *how to design insightful questions* based on the given label $y_i$, which is formulated as follows:

$$\Phi_i = \{(Q_i^j, A_i^j)\}_{j=1}^{N_q} := F_{vqg}(y_i) \tag{1}$$

where $y_i$ might be corrupted label $\tilde{y}_i$ or ground-truth label $\hat{y}_i$, $Q_i^j$ denotes the $j$-th question and $A_i^j$ denotes expected answer, and $N_q$ denotes the number of questions. In order to comprehensively and fully understand the semantics of images, two different types of questions are considered in VDC, including *coarse-grained general questions* and *fine-grained label-specific questions*.

**General Questions.** General questions can serve as a means to acquire holistic semantic understanding of an image from a global perspective, such as *"Please describe the image briefly."*. The expected answers to these general questions align with the given label. Since the general questions remain consistent across various labels, they are generated by random selection from a set of predefined templates, as outlined in Table 10 in Appendix E.

**Label-specific Questions.** Besides, the label-specific questions related to the given labels aim to extract more localized semantics from the image, encompassing aspects of common sense features, attributions, functions, geography, history, culture, and *etc* . For example, given the label "*airplane*", an apt question is "*Is the object in the image designed for flying in the air?*". Designing most label-specific questions necessitates a level of expertise about the label that may exceed the capacity of a human annotator. When dealing with a multitude of labels, such as ImageNet with 1,000 classes, manually designing for each label becomes impractical. Hence, we utilize LLM like ChatGPT (OpenAI) to automatically generate these questions, depending on its expansive open-world knowledge. The well-designed prompts and generated questions are detailed in Appendix D and E.

## 4.2 VISUAL QUESTION ANSWERING

The next step involves responding to the generated questions in Sec 4.1 based on the input image $\boldsymbol{x}_i$ to acquire the semantics of the visual content. This process is often referred to as the visual question answering (VQA) task, which can be formulated as follows:

$$R_i^j := F_{vqa}(Q_i^j, \boldsymbol{x}_i) \tag{2}$$

where $R_i^j$ indicates the response of VQA model for the question $Q_i^j$. Answering these questions necessitates the capabilities of natural language generation and external knowledge beyond the visible content of image. Therefore, we resort to MLLM as our VQA model owing to its remarkable

capabilities of visual and language understanding and reasoning, which has been demonstrated in a wide range of visual-language tasks.

## 4.3 VISUAL ANSWER EVALUATION

Afterward, for a suspicious input sample $(\boldsymbol{x}_i, y_i)$, we obtain a set of questions, expected answers, and responses, *i.e.*, $\{Q_i^j, A_i^j, R_i^j\}_{j=1}^{N_q}$. The subsequent step is to assess visual-linguistic consistency by evaluating the matching score between the semantics of the image and label. We first judge the correctness of the response of MLLM, *i.e.*, whether it aligns with the expected answer, which can be formulated as follows:

$$e_i^j := F_{vae}(A_i^j, R_i^j) \tag{3}$$

where $e_i$ denotes the correctness, *i.e.*, *true* or *false*. For label-specific questions with deterministic expected answers, we use string matching to evaluate the response. If the word "yes" is present in the response, the result should be true, otherwise if the response contains the word "no", the result should be false. Nevertheless, for general questions, string matching is insufficient to determine correctness. In such cases, we employ ChatGPT as a specialized evaluator through meticulously designed prompts, which is a commonly adopted approach in the evaluation of LLM Chang et al. (2023).

**Vote-based Ensemble.** Then the matching score $s_i$ of sample $(\boldsymbol{x}_i, y_i)$ is computed as the proportion of questions answered correctly, which are formulated as follows:

$$s_i = \frac{\sum_{j=1}^{N_q} \mathbb{1}(e_i = \textit{true})}{N_q} \tag{4}$$

where $\mathbb{1}(\cdot)$ denotes identity function. If the score is less than the threshold $\alpha$, sample $(\boldsymbol{x}_i, y_i)$ is detected as a dirty sample and then removed from the dataset.

## 5 EXPERIMENTS

### 5.1 EXPERIMENTAL SETTINGS

**Datasets.** We evaluate ASRon three benchmark datasets, CIFAR-10 (Krizhevsky et al., 2009) and two ImageNet (Russakovsky et al., 2015) subsets: (1) For ImageNet-100, we randomly choose 100 classes from ImageNet, in which 500 images per class for training and 100 images per class for testing. (2) For ImageNet-Dog, to evaluate the effect of similarity of classes, we randomly choose 10 classes of dogs from ImageNet, in which 800 images per class for training and 200 images per class for testing.

**Dirty Samples Generation.** Denote the ratio of dirty samples in the whole dataset by $\eta$. Two types of dirty samples are considered in the evaluation, which are illstrated as follows:

- **Poisoned Samples.** We consider six representative backdoor attacks to generate poisoned samples: (1) Visible triggers: BadNets (Gu et al., 2019), Blended (Chen et al., 2017), TrojanNN (Liu et al., 2018). (2) Invisible triggers: SIG (Barni et al., 2019), SSBA Li et al. (2021), WaNet Nguyen & Tran (2021). For all attacks, we randomly choose the same number of images from all classes except target class to add trigger, and then change the labels as target label. The example and settings of each attack are detailed in Appendix C.2.

- **Noisy Labels.** We experiment with two popular synthetic noisy model models: the symmetric and asymmetric noise: (1) Symmetric noisy label is generated by uniform flipping, *i.e.*, randomly flipping a ground-truth label to all other possible classes (Kim et al., 2019). (2) Asymmetric noisy label is generated by flipping the ground-truth label to the next class, *i.e.*, $(i \bmod K) + 1$, where $K$ denotes the number of classes.

**Evaluation Metrics.** We report the detection results with two key metrics: *true positive rate (TPR)* and *false positive rate (FPR)* following Qi et al. (2023). TPR means the recall of detected dirty samples, representing the capacity to successfully detect dirty samples within the dataset. FPR denotes the ratio of clean samples erroneously identified as dirty samples, highlighting the susceptibility to produce false alarms. An ideal detection method should exhibit a higher TPR and lower FPR. Let $v_i = 1$ indicate that the $i$-th sample is detected as dirty sample. Moreover, when retraining on

Table 1: Comparison of TPR (%) and FPR (%) for poisoned sample detection on CIFAR-10. $\eta = 0.09$, *i.e.*, 500 poisoned samples per class. Average is the mean of results of different triggers. Top 2 are **bold**.

| Method | Clean Data | BadNets TPR↑ | FPR↓ | Blended TPR↑ | FPR↓ | SIG TPR↑ | FPR↓ | TrojanNN TPR↑ | FPR↓ | SSBA TPR↑ | FPR↓ | WaNet TPR↑ | FPR↓ | Average TPR↑ | FPR↓ |
|---|---|---|---|---|---|---|---|---|---|---|---|---|---|---|---|
| | | | | | | **Dataset: CIFAR-10** | $\eta = 0.09$ | **(500 poisoned samples per class)** | | | | | | | |
| STRIP | 4% | 94.22 | 10.99 | 32.82 | 11.12 | **100.00** | 10.98 | 99.73 | 10.05 | 81.87 | 9.33 | 3.82 | 10.45 | 68.74 | 10.49 |
| SS | 4% | 61.62 | 48.85 | 61.40 | 48.87 | 60.89 | 48.92 | 59.53 | 49.06 | 58.02 | 49.21 | 57.22 | 49.29 | 59.78 | 49.03 |
| SCAn | 4% | 96.49 | 2.82 | 93.49 | 2.80 | 99.47 | **2.59** | 99.90 | 2.85 | 92.49 | 2.83 | 90.93 | 2.99 | 95.46 | 2.81 |
| Frequency | 4% | 88.98 | 18.71 | 82.80 | 18.70 | 48.07 | 20.79 | **100.00** | 11.40 | 85.84 | 19.81 | 40.02 | 20.61 | 74.29 | 18.34 |
| CT | 4% | **97.24** | **0.18** | **97.78** | **1.02** | 99.16 | **0.74** | **100.00** | **0.13** | **98.31** | **0.10** | 95.16 | **0.70** | **97.94** | **0.48** |
| D-BR | 0% | 87.13 | 3.36 | 23.93 | 7.60 | 94.40 | 2.56 | 80.85 | 10.28 | 10.07 | 8.93 | 10.18 | 8.87 | 51.09 | 6.93 |
| SPECTRE | 0% | 94.00 | 20.62 | 95.31 | 20.49 | 8.16 | 29.11 | 80.07 | 22.00 | **97.44** | 20.28 | 88.24 | 21.19 | 77.20 | 22.29 |
| VDC (Ours) | 0% | **99.93** | 2.75 | **99.87** | 2.75 | **99.84** | 2.75 | **99.93** | 2.75 | **99.91** | 2.75 | **99.96** | 2.75 | **99.91** | 2.75 |

Table 2: Comparison of TPR (%) and FPR (%) for poisoned sample detection on ImageNet-100. $\eta = 0.099$, *i.e.*, 50 poisoned samples per class. Average is the mean of results of different triggers. Top 2 are **bold**.

| Method | Clean Data | BadNets TPR↑ | FPR↓ | Blended TPR↑ | FPR↓ | SIG TPR↑ | FPR↓ | TrojanNN TPR↑ | FPR↓ | SSBA TPR↑ | FPR↓ | WaNet TPR↑ | FPR↓ | Average TPR↑ | FPR↓ |
|---|---|---|---|---|---|---|---|---|---|---|---|---|---|---|---|
| | | | | | | **Dataset: ImageNet-100** | $\eta = 0.099$ | **(50 poisoned samples per class)** | | | | | | | |
| STRIP | 4% | 92.20 | 12.56 | 99.19 | 11.79 | **100.00** | 10.95 | **100.00** | 13.14 | **99.74** | 11.26 | 3.11 | 11.32 | **82.37** | 11.84 |
| SS | 4% | 48.12 | 50.21 | 44.95 | 50.55 | 44.95 | 50.55 | 44.95 | 50.55 | 44.95 | 50.55 | 45.13 | 50.53 | 45.51 | 50.49 |
| SCAn | 4% | **97.01** | 1.66 | 97.58 | 2.46 | 99.21 | **1.21** | 99.92 | 1.77 | 87.39 | **1.36** | **58.97** | 2.54 | 90.01 | 1.83 |
| Frequency | 4% | 1.52 | 1.59 | 1.31 | 1.59 | 1.72 | 1.59 | 95.05 | 1.59 | 3.41 | 1.59 | 0.04 | 1.59 | 17.18 | 1.59 |
| CT | 4% | 94.16 | **0.37** | **99.21** | **0.37** | 99.35 | **0.06** | 99.84 | **0.58** | 91.47 | **0.39** | 0.00 | **0.69** | 80.67 | **0.41** |
| D-BR | 0% | 86.43 | 23.56 | 9.56 | 10.03 | 76.09 | 15.07 | 16.53 | 9.06 | 11.09 | 10.15 | 10.08 | 9.87 | 34.96 | 12.96 |
| SPECTRE | 0% | 48.57 | 50.16 | 44.95 | 50.55 | 44.95 | 50.55 | 44.97 | 50.55 | 44.95 | 50.55 | 45.09 | 50.54 | 45.58 | 50.48 |
| VDC (Ours) | 0% | **99.92** | 1.55 | **99.94** | 1.55 | **99.90** | 1.55 | **99.96** | 1.55 | **99.98** | 1.55 | **99.94** | 1.55 | **99.94** | 1.55 |

the purified dataset, we report the attack success rate (ASR) and the clean accuracy (ACC) of the retrained model.

**Implemented Details.** We adopt ChatGPT based on GPT-3.5-turbo (OpenAI) as LLM and Instruct-BLIP (Dai et al., 2023) as MLLM in VDC. For all datasets, we generate two general questions. The number of label-specific questions is six for ImageNet-100 and four for CIFAR-10 and ImageNet-Dog. The threshold $\alpha$ is set as $0.5$ across all experiments. The noisy ratio $\eta$ for noisy labels is set as $0.4$. We poison 50 and 500 samples per class for CIFAR-10, 5 and 50 per class for ImageNet-100, and 80 per class for ImageNet-Dog. We retrain on the purified dataset with ResNet-18 (He et al., 2016). Additional details can be found in Appendix C.1.

**Compared Baselines.** For poisoned sample detection, we compare with 7 baselines, in which STRIP (Gao et al., 2019), SS (Tran et al., 2018), SCAn (Tang et al., 2021), Frequency (Zeng et al., 2021) and CT (Qi et al., 2023) require external clean subset to execute, while SPECTRE (Hayase et al., 2021) and D-BR (Chen et al., 2022) do not require any clean subset. For noisy label detection, we compare with 5 baselines, including BHN (Yu et al., 2023), CL (Northcutt et al., 2021), CORES Cheng et al. (2021), SimiFeat-V and SimiFeat-R (Zhu et al., 2022), in which BHN relies on a clean subset to perform. The detailed settings of each baseline can be found in Appendix C.3,C.4.

## 5.2 EXPERIMENTAL RESULTS

### 5.2.1 RESULTS ON DETECTING POISONED SAMPLES

In this section, we first conduct a comprehensive evaluation on the poisoned samples detection. The results on CIFAR-10, ImageNet-100 and ImageNet-Dog with different poisoning ratios are presented in Tables 1,2,13,14 (Refer Tables 13,14 in Appendix F.2). For a fair comparison, all baselines requiring clean data utilize 4% clean subset. The results demonstrate the effectiveness of our proposed method from the following aspects:

**Consistent Effectiveness Against Various Types of Poisoned Samples.** From the results on CIFAR-10 in Table 1, we find that VDC consistently exhibits superior performance under various types of poisoned samples without relying on any clean subset, demonstrating the generalization of VDC. In contrast, other detectors are sensitive to different types of triggers. For example, VDC achieves average TPR of $99.91\%$ against all backdoor attacks, while SPECTER experiences a sig-

Table 3: Comparison of TPR (%) and FPR (%) for poisoned sample detection on ImageNet-Dog. $\eta = 0.09$, *i.e.*, 80 poisoned samples per class. Average is the mean of results of different triggers. Top 2 are **bold**.

| Method | Clean Data | BadNets TPR↑ | FPR↓ | Blended TPR↑ | FPR↓ | SIG TPR↑ | FPR↓ | TrojanNN TPR↑ | FPR↓ | SSBA TPR↑ | FPR↓ | WaNet TPR↑ | FPR↓ | Average TPR↑ | FPR↓ |
|---|---|---|---|---|---|---|---|---|---|---|---|---|---|---|---|
| Strip | 4% | 93.19 | 10.88 | 82.78 | 11.18 | 97.08 | 11.95 | **98.61** | 11.32 | 95.42 | 10.62 | 3.47 | 9.42 | 78.43 | 10.90 |
| SS | 4% | 19.86 | 21.66 | 17.64 | 21.88 | 21.94 | 21.46 | 23.89 | 21.26 | 23.75 | 21.28 | 12.50 | 22.39 | 19.93 | 21.66 |
| SCAn | 4% | 98.06 | 0.01 | 72.36 | 9.15 | 78.61 | **0.03** | 62.22 | **0.23** | 84.44 | **0.15** | 12.54 | **2.12** | 68.04 | **1.95** |
| Frequency | 4% | 83.89 | 45.48 | 50.00 | 45.45 | 44.03 | 45.48 | 95.97 | 44.64 | 61.94 | 45.45 | 36.53 | 45.65 | 62.06 | 45.36 |
| CT | 4% | 92.50 | **0.99** | 84.31 | **0.58** | 15.41 | 0.99 | 98.06 | 0.44 | 88.89 | 0.29 | 0.00 | 0.92 | 63.20 | 0.70 |
| D-BR | 0% | 8.61 | 8.85 | 9.31 | 8.85 | 10.83 | 9.22 | 9.72 | 9.01 | 8.75 | 9.15 | 8.47 | 9.12 | 9.28 | 9.03 |
| SPECTRE | 0% | **99.44** | 45.11 | 77.64 | 47.27 | **99.86** | 45.07 | 97.50 | 45.30 | 96.94 | 45.36 | **53.19** | 49.68 | **87.43** | 46.30 |
| VDC (Ours) | 0% | **98.89** | 4.12 | **97.50** | 4.12 | **98.61** | 4.12 | **99.31** | 4.12 | **98.89** | 4.12 | **98.89** | 4.12 | **98.68** | 4.12 |

Table 4: Comparison of TPR (%) and FPR (%) for noisy label detection on the three datasets under different types of noisy labels, where noisy ratio $\eta = 0.4$. Top 2 are **bold**.

| Method | Clean Data | CIFAR-10 $\eta = 0.4$ Symmetric TPR↑ | FPR↓ | Asymmetric TPR↑ | FPR↓ | ImageNet-100 $\eta = 0.4$ Symmetric TPR↑ | FPR↓ | Asymmetric TPR↑ | FPR↓ | ImageNet-Dog $\eta = 0.4$ Symmetric TPR↑ | FPR↓ | Asymmetric TPR↑ | FPR↓ |
|---|---|---|---|---|---|---|---|---|---|---|---|---|---|
| BHN | 20% | 80.88 | **2.98** | **83.13** | **3.24** | 57.04 | **1.94** | 16.24 | **0.96** | 14.36 | **0.23** | 22.54 | **0.75** |
| CORES | 0% | 92.11 | 4.85 | 5.36 | 4.47 | 77.22 | 2.06 | 0.05 | **0.07** | 84.44 | 23.87 | 44.04 | 24.47 |
| CL | 0% | 85.05 | 8.75 | 82.49 | 4.50 | 67.32 | 19.07 | 43.62 | 17.82 | 90.78 | 71.37 | 61.97 | 46.86 |
| SimiFeat-V | 0% | 98.80 | 4.13 | 59.67 | 7.43 | **98.31** | 5.52 | 55.67 | 17.65 | 89.59 | 11.73 | 51.85 | 22.10 |
| SimiFeat-R | 0% | **99.16** | 5.11 | 79.46 | 15.18 | **99.27** | 8.22 | **69.59** | 27.25 | **95.86** | 17.90 | **66.39** | 35.07 |
| VDC (Ours) | 0% | **98.81** | **2.61** | **99.60** | **2.62** | 94.79 | **1.55** | **92.34** | 1.55 | **97.30** | **7.90** | **91.97** | **7.90** |

nificant fluctuation with a difference of $87.15\%$ between its highest and lowest TPR. Additionally, VDC achieves competitive results in terms of FPR, averaging only $2.75\%$, which indicates that VDC has a low propensity to incorrectly identify clean samples as dirty samples.

**Consistent Effectiveness Across Datasets.** Comparing the results of ImageNet-100 in Table 2 and CIFAR-10 in Table 1, when facing a larger dataset with more labels, VDC maintains performance with the average TPR still reaching $99.94\%$. On the contrary, other baselines are unstable on different datasets, such as SPECTRE decreases from $77.20\%$ to $45.58\%$. To explore the effect of the similarity of classes, we evaluate on a fine-grained dataset ImageNet-Dog. From the results in Table 3 in Appendix F.1, VDC shows evident improvement compared to other baselines.

**Consistent Effectiveness Across Poisoning Ratios.** We also evaluate with lower poisoning ratios on CIFAR-10 and ImageNet-100 to study the effect of poisoning ratios. Compare Table 1 with $\eta = 0.09$ and Table 13 with $\eta = 0.009$ on CIFAR-10, we find that the performance of VDC has almost no fluctuation, while other methods are greatly affected by the poisoning ratio. A similar phenomenon on ImageNet-100 can be found in Table 2 and 14.

### 5.2.2 RESULTS ON DETECTING NOISY LABELS.

In this section, we evaluate VDC on noisy label detection, another common type of dirty samples. The results on CIFAR-10, ImageNet-100 and ImageNet-Dog are shown in Table 4, verifying that VDC also performs well on detecting noisy labels from the following points:

**Consistent Effectiveness Against Various Types of Noisy Labels.** By comparing the performance on the symmetric and the asymmetric noisy labels, we note that asymmetric is a more challenging setting. Even though some baselines behave well on detecting symmetric noisy labels, such as SimiFeat-V and SimiFeat-R, they may reach low TPR on the symmetric noisy labels. However, VDC consistently works well on the asymmetric noisy label. For example, VDC achieves $99.60\%$ TPR on detecting asymmetric noisy labels on CIFAR-10, while SimiFeat-V only has $59.67\%$ TPR.

**Consistent Effectiveness Across Datasets.** From the results on the three datasets in Table 4, we note that VDC performs consistently well on different datasets, while other methods perform worse on ImageNet-100 and Imagenet-Dog, which indicates the robustness of our proposed method.

Table 5: Comparison of TPR (%) and FPR (%) for detecting the mixture of poisoned sampels and noisy labels on CIFAR-10, where poisoning ratio $\eta_1$ and noisy ratio $\eta_2$ are set as 0.1. Top 2 are **bold**.

| | | Dataset: CIFAR-10 | | | | poisoning ratio $\eta_1 = 0.09$ | | | noisy ratio $\eta_2 = 0.1$ | | | | | | |
|---|---|---|---|---|---|---|---|---|---|---|---|---|---|---|---|
| Method | Clean Data | BadNets | | Blended | | SIG | | TrojanNN | | SSBA | | WaNet | | Average | |
| | | TPR↑ | FPR↓ | TPR↑ | FPR↓ | TPR↑ | FPR↓ | TPR↑ | FPR↓ | TPR↑ | FPR↓ | TPR↑ | FPR↓ | TPR↑ | FPR↓ |
| STRIP | 4% | 50.59 | 12.41 | 27.40 | 11.74 | 50.83 | 12.09 | 50.31 | 12.72 | 43.54 | 9.39 | 4.98 | 11.64 | 37.94 | 11.67 |
| SS | 4% | 51.63 | 49.61 | 52.77 | 49.34 | 53.80 | 49.10 | 50.91 | 49.78 | 51.45 | 49.65 | 50.57 | 49.86 | 51.86 | 49.56 |
| SCAn | 4% | 45.38 | **0.00** | 45.64 | 3.80 | 50.71 | 1.77 | 47.46 | **0.00** | 44.67 | **0.01** | 43.89 | 4.07 | 46.29 | 1.61 |
| Frequency | 4% | 52.22 | 18.65 | 49.21 | 18.66 | 34.08 | 20.71 | 53.18 | 11.44 | 51.44 | 19.73 | 30.14 | 20.53 | 45.05 | 18.29 |
| CT | 4% | 45.22 | 0.32 | 47.88 | **0.39** | 48.37 | **0.22** | 49.41 | 1.61 | 46.67 | 0.38 | 46.00 | **0.69** | 47.26 | **0.60** |
| D-BR | 0% | 31.92 | **0.00** | 12.09 | 1.07 | 1.20 | 1.42 | 16.13 | 17.73 | 0.01 | **0.02** | 3.08 | 3.20 | 10.74 | 3.91 |
| SPECTRE | 0% | 23.42 | 22.07 | 22.49 | 22.29 | 26.09 | 27.51 | 22.41 | 22.32 | 21.67 | 22.49 | 34.22 | 23.96 | 25.05 | 23.44 |
| BHN | 20% | 68.40 | 1.27 | 69.19 | 1.34 | **70.35** | 1.35 | **72.61** | 1.12 | 67.81 | 1.26 | 69.43 | 1.34 | 69.63 | 1.28 |
| CL | 0% | 49.69 | 0.80 | 33.11 | **0.70** | 34.32 | **0.53** | 33.21 | **0.51** | 33.85 | 0.67 | 33.77 | **0.74** | 36.33 | **0.66** |
| CORES | 0% | 66.73 | 2.29 | 47.59 | 2.46 | 30.41 | 15.94 | 47.27 | 2.18 | 48.03 | 2.55 | 49.76 | 2.92 | 48.30 | 4.72 |
| SimiFeat-V | 0% | 80.02 | 4.71 | 77.94 | 5.06 | 66.72 | 5.31 | 52.60 | 4.89 | **85.73** | 4.72 | 87.48 | 4.80 | **75.08** | 4.92 |
| SimiFeat-R | 0% | **81.36** | 4.57 | **79.12** | 5.48 | 66.23 | 6.17 | 52.42 | 4.93 | 80.85 | 5.24 | **89.19** | 4.93 | 74.86 | 5.22 |
| VDC (Ours) | 0% | **99.42** | 2.79 | **99.40** | 2.79 | **99.39** | 2.79 | **99.42** | 2.79 | **99.41** | 2.79 | **99.43** | 2.79 | **99.41** | 2.79 |

Table 6: Comparison of ASR (%) and ACC (%) for training on the purified CIFAR-10 with poisoning ratio $\eta = 0.09$. Top 2 are **bold**.

| Method | BadNets | | Blended | | SIG | | TrojanNN | | SSBA | | WaNet | | Average | |
|---|---|---|---|---|---|---|---|---|---|---|---|---|---|---|
| | ASR↓ | ACC↑ | ASR↓ | ACC↑ | ASR↓ | ACC↑ | ASR↓ | ACC↑ | ASR↓ | ACC↑ | ASR↓ | ACC↑ | ASR↓ | ACC↑ |
| No detection | 96.31 | 92.33 | 98.16 | 93.30 | 99.99 | 93.51 | 100.00 | 93.43 | 98.16 | 92.66 | 95.38 | 92.89 | 98.00 | 93.02 |
| Strip | 1.82 | 92.38 | 98.18 | 92.9 | **0.23** | 93.15 | 81.38 | **93.61** | 79.69 | 92.26 | 94.68 | 92.95 | 59.33 | 92.88 |
| SS | 89.84 | 86.43 | 94.49 | 87.59 | 99.97 | 86.09 | 99.79 | 88.17 | 96.79 | 87.2 | 80.77 | 85.19 | 93.61 | 86.78 |
| SCAn | 0.97 | **93.39** | 32.21 | 93.53 | 20.11 | 92.5 | 7.51 | 93.34 | 17.67 | 92.55 | 5.17 | 93.04 | 13.94 | 93.06 |
| Frequency | 75.71 | 92.05 | 87.3 | 91.9 | 99.89 | 91.95 | 1.27 | 93.12 | 66.58 | 90.2 | 91.88 | 91.6 | 70.44 | 91.80 |
| CT | **0.82** | 92.83 | **1.93** | 93.4 | 1.26 | 92.91 | **2.31** | 93.2 | **3.19** | **93.09** | **2.36** | **93.09** | **1.98** | **93.09** |
| D-BR | 88.14 | 93.23 | 95.78 | 91.44 | 99.97 | **93.82** | 100 | 93.21 | 97.42 | 92.54 | 95.54 | 92.33 | 96.14 | 92.76 |
| SPECTRE | 71.89 | 87.92 | 45.57 | 87.6 | 99.9 | 86.76 | 97.6 | 87.76 | 4.77 | 88.31 | 18.53 | 88.19 | 56.38 | 87.76 |
| VDC (Ours) | **0.86** | **93.32** | **1.23** | 92.24 | **1.24** | 93.15 | 4.41 | **93.88** | **1.12** | **93.11** | **0.94** | **93.57** | **1.63** | **93.21** |

### 5.2.3 Results on Detecting Hybrid Dirty Samples

In the real world, when an attacker poisons a realistic dataset, the dataset may already contain noisy labels. Therefore, in this section, we further evaluate the effectiveness of detectors when the dataset contains both poisoned samples and noisy samples, in which poisoning ratio is 0.09 and noisy ratio is 0.1 The results on CIFAR-10 are shown in Table 5. The following insightful points can be found from the results:

**Consistent Effectiveness Against Hybrids of Poisoned Samples and Noisy Labels.** In this more challenging scenario, VDC still shows leading advantages compared with other methods, with average TPR reaching 99.41%. However, methods designed only for poisoned sample detection perform poorly when detecting a mixture of various dirty samples, such as SCAn decreasing from 95.46% to 46.29%. In the meantime, methods designed only to detect noisy samples also underperform in this case, such as CL decreasing from 85.05% to 36.33%, which further illustrates the effectiveness and robustness of our proposed method.

### 5.2.4 Training on the Purified Datasets

After detecting and removing dirty samples from the origin dataset, we normally train DNNs on the purified datasets to verify the detection effect. The results on the purified datasets initially contain poisoned samples, noisy labels, and hybrid dirty samples are shown in Tables 6,15,16,17. By accurately detecting dirty samples, VDC indeed prevents the trained model from being interfered by dirty samples, *i.e.*, maintaining low ASR and high ACC compared with other detectors.

## 6 A CLOSER LOOK AT VDC

In this section, we provide further analysis and ablation studies of VDC and show some limitations.

**Effect of the Type of Visual Questions.** Figure 2a illustrates the influence of visual question types generated in VDC. We conducted experiments separately only using general questions or label-

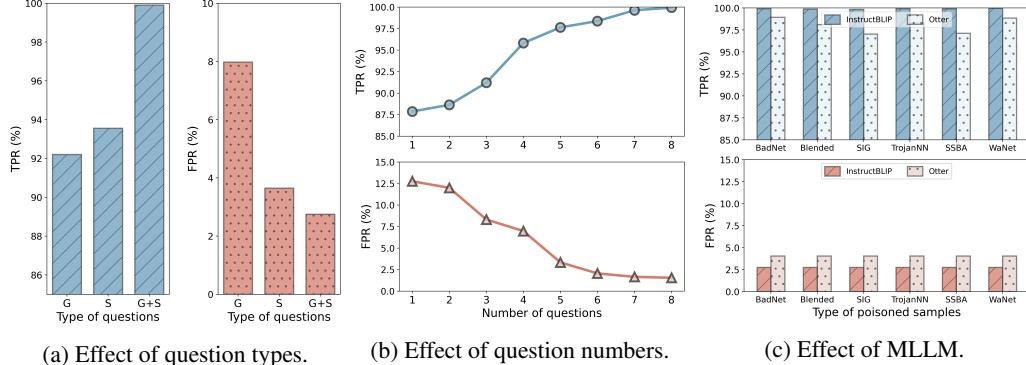

Figure 2: Ablation results on the different aspects of VDC. (a) shows average results on CIFAR-10 with $\eta = 0.09$ under different types of visual questions, where G denotes general questions and S denotes label-specific questions. (b) shows average results on ImageNet-100 with $\eta = 0.099$ under different numbers of visual questions. (c) shows results of various poisoned samples on CIFAR-10 with $\eta = 0.09$ under different multimodal large language models.

specific questions while keeping all other settings constant. We observe that using only one type of question makes the model perform worse. In addition, label-specific questions are slightly more important than general questions.

**Effect of the Number of Visual Questions.** We investigate the effect of the number of visual questions generated in VDC. Figure 2b shows the detection results *w.r.t.* various number of questions. We find that VDC's performance improves as the number of questions increases. But more questions also lead to more inference time. Therefore, it becomes crucial to strike a balance between these two factors.

**Effect of the Multimodal Large Language Model.** In Figure 2c, we substitute the multimodal large language model in the VDC with Otter (Li et al., 2023), another recently open-sourced MLLM, to investigate the impact of MLLM. Although the performance differs from those obtained with InstructBLIP, it still outperforms the majority of baselines. with the TPR for all poisoned samples consistently exceeding $96\%$, which further verifies the effectiveness of VDC.

**Computational Complexity.** Unlike other baselines that require training, VDC requires only inference of LLM and MLLM. Let $K$ represent the number of classes, $N_{q_g}$ and $N_{q_s}$ denote the number of general questions and label-specific questions respectively, $T$ and $T'$ denote the time of one inference of LLM and MLLM. The overall time complexity can be expressed as $O(TKN_{q_s}) + O(T'(N_{q_g} + N_{q_s})N) + O(TN_{q_g}N)$, in which three terms correspond to the complexities of VQG, VQA, and VQE respectively. With the development of lightweight LLM, such as quantization (Yao et al., 2023), the inference speed of LLM will increase, leading to a further reduction in the computational cost of VDC.

**Limitations. 1)** VDC hinges on the inconsistency between visual content and labels, making it inappropriate for detecting samples without corrupted labels, such as clean-label backdoor attack. **2)** Although ensembling technique has been employed in our framework to mitigate the risk of abnormal questions and answers, LLM and MLLM may still yield incorrect replies. However, the performance of VDC will also improve in the future as LLM progresses.

## 7 CONCLUSION

In this paper, we propose to detect dirty samples with corrupted labels by exploiting semantic inconsistency between visual content and associated labels. To this end, we design versatile data cleanser (VDC), a universal detection framework harnessing the surpassing capabilities of large language models and multimodal large language models, which is capable of detecting various categories and types of dirty samples. Experimental results validate the consistent superior performance of VDC in poisoned sample detection and noisy label detection. In addtion, VDC still maintains effectiveness even when the dataset contains the hybrid dirty samples. Furthermore, we anticipate that as large language models continue to evolve at a rapid pace, VDC will demonstrate further enhanced performance in the future.

## ACKNOWLEDGMENTS

This work was supported by the National Natural Science Foundation of China under grant No. 62076213, Shenzhen Science and Technology Program under grants No. RCYX20210609103057050, Outstanding Youth Program of Guangdong Natural Science Foundation, and the Guangdong Provincial Key Laboratory of Big Data Computing, the Chinese University of Hong Kong, Shenzhen.

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

# A APPENDIX OVERVIEW

The overall structure of the Appendix is listed as follows:

# B A NAIVE APPROACH WITH CLIP

As we identified in the manuscript, *how to measure the visual-linguistic inconsistency between the visual content and associated labels* is the key to detect dirty samples. A naive approach to quantify such semantic inconsistency is directly using CLIP (Radford et al., 2021). We first encode the input image using image encoder in the CLIP, and get the image representation $\mathbf{I}$. The associated label is transformed into sentences, "a photo of {label}". Then the text representation $\mathbf{T}$ is extracted from the sentence via text encoder in the CLIP. Cosine similarity between $\mathbf{I}$ and $\mathbf{T}$ is treated as the matching score. If the matching score is less than a certain threshold, the input sample can be considered as dirty sample. In the implementation, we choose ViT-B/32 as the image encoder and the threshold is set as 0.2. The results are shown in Table 7, where VDC-CLIP represents the naive approach with CLIP. We find that the TPR using only CLIP is far from our proposed VDC, indicating the need for more advanced detection frameworks instead of only using CLIP.

Table 7: Comparsion of TPR (%) and FPR (%) on the poisoned sample detection. VDC-CLIP denotes the naive approach with CLIP.

| Dataset | Method | BadNets | | Blended | | SIG | | TrojanNN | | SSBA | | WaNet | |
|---|---|---|---|---|---|---|---|---|---|---|---|---|---|
| | | TPR | FPR | TPR | FPR | TPR | FPR | TPR | FPR | TPR | FPR | TPR | FPR |
| CIFAR-10 $\eta = 0.09$ | VDC-CLIP | 41.89 | 1.98 | 48.02 | 1.98 | 28.37 | 1.98 | 38.07 | 1.98 | 51.95 | 1.98 | 34.60 | 1.98 |
| | VDC (Ours) | 99.93 | 2.75 | 99.87 | 2.75 | 99.84 | 2.75 | 99.93 | 2.75 | 99.91 | 2.75 | 99.96 | 2.75 |
| CIFAR-10 $\eta = 0.009$ | VDC-CLIP | 41.55 | 2.83 | 49.33 | 2.83 | 29.33 | 2.83 | 39.56 | 2.83 | 52.00 | 2.83 | 36.78 | 2.83 |
| | VDC (Ours) | 100.00 | 2.72 | 99.56 | 2.72 | 99.78 | 2.72 | 100.00 | 2.72 | 99.78 | 2.72 | 100.00 | 2.72 |
| ImageNet-100 $\eta = 0.099$ | VDC-CLIP | 80.81 | 1.84 | 77.52 | 1.84 | 82.34 | 1.84 | 70.14 | 1.84 | 77.90 | 1.84 | 82.20 | 1.84 |
| | VDC (Ours) | 99.92 | 1.55 | 99.94 | 1.55 | 99.90 | 1.55 | 99.96 | 1.55 | 99.98 | 1.55 | 99.94 | 1.55 |
| ImageNet-100 $\eta = 0.0099$ | VDC-CLIP | 81.40 | 1.75 | 77.00 | 1.75 | 82.60 | 1.75 | 71.11 | 1.75 | 77.78 | 1.75 | 83.23 | 1.75 |
| | VDC (Ours) | 99.80 | 1.55 | 100.00 | 1.55 | 99.80 | 1.55 | 100.00 | 1.55 | 100.00 | 1.55 | 99.80 | 1.55 |
| ImageNet-Dog $\eta = 0.09$ | VDC-CLIP | 12.50 | 3.23 | 4.44 | 3.23 | 7.22 | 3.23 | 11.94 | 3.23 | 4.58 | 3.23 | 9.58 | 3.23 |
| | VDC (Ours) | 98.89 | 4.12 | 97.50 | 4.12 | 98.61 | 4.12 | 99.31 | 4.12 | 98.89 | 4.12 | 98.89 | 4.12 |

# C MORE IMPLEMENTATION DETAILS

## C.1 DETAILS OF TRAINING ON THE PURIFIED DATASETS

After successfully detecting dirty samples in the dataset, we need to normally training on the purified dataset t further verify the effectiveness of detectors. In our experiments, we choose ResNet-18 as the target model. For all datasets, the training epochs is set as 100 and adopt SGD optimizer. For

CIFAR-10, we set the batch size of 128 and the inital learning rate of 0.1 and decreases it by the factor of 10 after 50, 75 epochs. For ImageNet-100 and ImageNet-Dog, the batch size is 64, the inital learning rate is 0.1 and decreases by the factor of 10 after 30, 60 epochs.

### C.2 DETAILS OF POISONED SAMPLE GENERATION

In this section, we present the settings for generating poisoned samples in backdoor attacks that are evaluated in the main manuscript. For all backdoor attacks, we choose class 0 as the target label.

**BadNets**   BadNets (Gu et al., 2019) stands as a seminal work in the realm of backdoor attacks, which introduces the concept of substituting specific pixels within a clean image with a well-designed trigger, thus yielding a poisoned image. In our experiments, for a $32 \times 32$ image in CIFAR-10, we select a $3 \times 3$ white square patch located in the lower-right corner of the image to serve as the trigger. In the case of images with dimensions $224 \times 224$ from both ImageNet-100 and ImageNet-Dog datasets, we utilize a white square patch with dimensions $21 \times 21$ as the trigger.

**Blended**   Blended Chen et al. (2017) firstly adopted the blended injection strategy to generate poisoned samples by blending a benign input instance with the key pattern. The choice of the key pattern can be an arbitrary image. In our experiments, we use a "Hello Kitty" cartoon image (see Figure 3) as a trigger, and the blending ratio is set as 0.1.

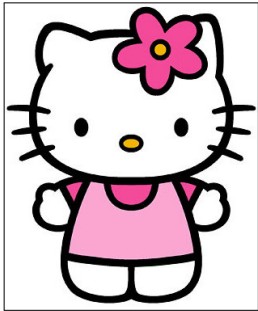

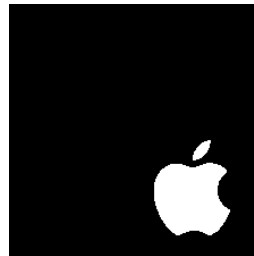

Figure 3: The Hello Kitty pattern used in Blended.

Figure 4: The trigger mask used in TrojanNN.

**SIG**   SIG (Barni et al., 2019) proposes a horizontal sinusoidal signal designed by $v(i,j) = \Delta \sin(2\pi j f/m), 1 \leq j \leq m, 1 \leq i \leq l$, for a certain frequency $f$, on the clean image, where $m$ is the number of columns of the image and $l$ the number of rows. In the evaluation, we set $\Delta = 20, f = 6$ for all datasets. The overlay backdooor signal is applied on all the channels. In this case, the backdoor is almost, though not perfectly, invisible.

**TrojanNN**   TrojanNN attack Liu et al. (2018) starts by choosing a trigger mask, which is a subset of the input variables that are used to inject the trigger. Then it searches for value assignment of the input variables in the trigger mask so that the selected neuron(s) of the target model can achieve the maximum values. The identified input values are essentially the trigger. In our evaluation, as shown in Figure 4, we choose to use the Apple logo as the trigger mask and ResNet-18 as target model.

**SSBA**   SSBA (Li et al., 2021) generates sample-specific invisible additive noises as backdoor triggers by encoding an attacker-specified string into clean images through an encoder-decoder network. Following the settings in (Li et al., 2021), we use a U-Net (Ronneberger et al., 2015) style DNN as the encoder, a spatial transformer network (Jaderberg et al., 2015) as the decoder. The encoder-decoder is trained for 140,000 iterations and batch size is set as 16.

**WaNet**   WaNet (Nguyen & Tran, 2021) uses a small and smooth warping field in generating poisoned images, making the modification unnoticeable. In our experiments, we adopt elastic image warping proposed in (Nguyen & Tran, 2021).

**Examples of Various Poisoned Samples** As shown in Figure 5, we choose one image from ImageNet-100 and visualize the examples of various poisoned samples mentioned above.

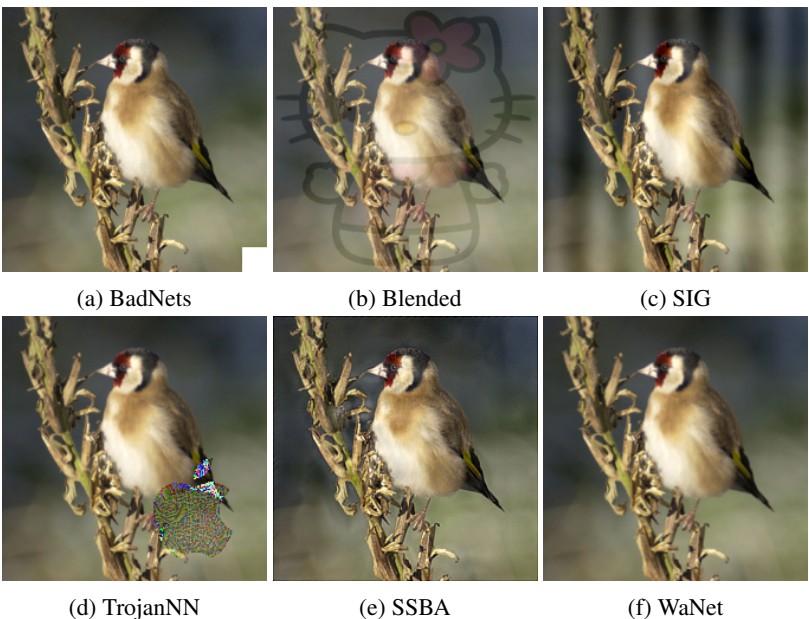

| (a) BadNets | (b) Blended | (c) SIG |
|---|---|---|
| (d) TrojanNN | (e) SSBA | (f) WaNet |

Figure 5: Examples of various types of poisoned samples.

### C.3 DETAILS OF BASELINE POISONED SAMPLE DETECTORS

In this section, we present the settings of 7 poisoned sample detection baselines compared in our experiments.

**STRIP** STRIP (Gao et al., 2019) detects a poisoned sample by checking whether superimposing the input image over a set of randomly selected images makes those new image's class label harder to predict. If so, the input is considered to be normal and otherwise. In our evaluation, the FRR is preset to be 0.1

**SS** SS (Tran et al., 2018) identifies spectral signatures of all known backdoor attacks to utilize tools from robust statistics to thwart the attacks. The upper bound on number of poisoned training set examples $\varepsilon$ is set as 0.1.

**SCAn** SCAn (Tang et al., 2021) utilizes several statistical methods to estimate the most likely parameters for the decomposition and untangling models and then detect an infected label through a likelihood ratio test. The threshold user for split clean samples in each classes is set as Euler's number $e$.

**Frequency** Frequency-based detection (Zeng et al., 2021) trains a binary classifier based on a training set that contains DCT transformations of clean samples and samples with digital manipulations. For CIFAR-10, We directly use their provided pretrained detection model. For ImageNet-100 and ImageNet-Dog, we train a 6 layer CNN with the same settings as CIFAR-10.

**CT** CT (Qi et al., 2023) proposes confusion training that applies an additional poisoning attack to the already poisoned dataset, actively decoupling benign correlation while exposing backdoor patterns to detection. In our experiments, we set confusion factor $\lambda = 20$, the number of confusion iterations $m = 6000$, the number of confusion training rounds $K = 6$.

**D-BR**  We only use the sample-distinguishment (SD) module in D-BR. SD module splits the whole training set into clean, poisoned and uncertain samples, according to the FCT metric. In our evaluation, we set $\alpha_c = 0.2$, $\alpha_p$ is set as the true poisoning ratio.

**SPECTRE**  SPECTRE (Hayase et al., 2021) uses robust covariance estimation to amplify the spectral signature of corrupted data. In our experiments, $\alpha$ is set as 4, poison fraction $\varepsilon$ is set as 0.1.

### C.4 DETAILS OF BASELINE NOISY LABEL DETECTORS

In this section, we present the settings of 5 noisy label detection baselines compared in our experiments.

**BHN**  BHN (Yu et al., 2023) defines the p-values based on the neural network with the clean data. The p-values are then applied to the multiple hypothesis testing to detect corrupted examples. In our evaluation, we set leave ratio as $0.4$. We use ResNet-18 for all datasets, and training epochs is set to be 200.

**CORES**  CORES (Cheng et al., 2021) trains ResNet-34 on the noisy dataset and uses its proposed sample sieve to filter out the corrupted examples. In our experiments, we adopt its default setting during training and calculate the F1 of the sieved out corrupted examples. The training epochs is set as 40.

**CL**  CL (Northcutt et al., 2021) detects corrupted labels by firstly estimating probabilistic thresholds to characterize the label noise, ranking examples based on model predictions, then filtering out corrupted examples based on ranking and thresholds.In our experiments, we train ResNet-18 on the noisy dataset and call the functions of Cleanlab[1] to detect noisy labels.

**SimiFeat-V and SimiFeat-R**  SimiFeat-V (Zhu et al., 2022) uses "local voting" via checking the noisy label consensuses of nearby features to determine if the example is corrupted. SimiFeat-R (Zhu et al., 2022) scores and ranks each instance based on the neighborhood information and filters out a guaranteed number of instances that are likely to be corrupted. In the evaluation, the KNN paprameter $k$ is set as 10 and epochs is set as 21.

## D  PROMPTS USED IN CHATGPT

In this section, we present the prompts that we used to query ChatGPT in our paper. Table 8 shows the prompts used for the generation of label-specific visual questions for different datasets. Table 9 shows the prompts used for the evaluation of the response of MLLM.

## E  EXAMPLES OF GENERATED QUESTIONS

In this section, we show some examples of generated visual questions in the visual question generation module of VDC.

### E.1 EXAMPLES OF GENERAL QUESTIONS

Table 10 shows the general questions used for acquiring holistic descriptions of the image, with some prompts sourced from (Liu et al., 2023).

### E.2 EXAMPLES OF LABEL-SPECIFIC QUESTIONS

Table 11 and 12 show the examples of label-specific visual questions on ImageNet-100.

---

[1] https://github.com/cleanlab/cleanlab

Table 8: The prompts used for the generation of label-specific visual questions for different datasets with ChatGPT. {**label**$_i$} represents the label name of class $i$, {**n**} denotes the number questions of each lable.

---

Dataset: **CIFAR-10, ImageNet-100**

Prompt: Please generate some visual questions to ask a multimodal large language model to identify if the label of an image is correct. These questions will help determine if the object in the image corresponds to the given label. Remember that the goal is to ask questions that would lead to a 'yes' answer if the label is correct.
The labels are ['{**label**$_1$}',···,'{**label**$_k$}'], generate {**n**} the most insightful questions for each label.
For example, if the label is 'airplane', the possible questions could be:
Can the object in the image be used to fly in the air?

---

Dataset: **ImageNet-Dog**

Prompt: Please generate some visual questions to ask a multimodal large language model to identify if the label of an image is correct. These questions will help determine if the breed of the dog in the image corresponds to the given label. Remember that the goal is to ask visual questions that would lead to a 'yes' answer if the label is correct.
The labels are ['{**label**$_1$}',···,'{**label**$_k$}'], generate {**n**} different questions for each label, such as the breed, attributes. The questions of each label should be used to judge different breeds.
For example, if the label is 'Chihuahua', the possible questions could be:
Does the dog in the image have any distinct features of a Chihuahua?

---

Table 9: The prompts used for the evaluation of the response of MLLM with ChatGPT. {**label**} represents the label name, {**response**} represents the response of MLLM in visual qunestion answering module.

---

Prompt: Assume you are a helpful and precise assistant for evaluation. Please judge whether the 'Caption' of an image and one of the 'Label' refer to the same object. Answer with yes or no.
- Caption: '{**response**}'
- Label: '{**label**}'

---

Table 10: The list of general questions for image description.

- Describe the image in detail.
- Describe the image briefly.
- How would you summarize the content of the image in a few words?
- Provide a detailed description of the given image.
- Describe the image concisely.
- Provide a brief description of the given image.
- Offer a succinct explanation of the picture presented.
- Summarize the visual content of the image.
- Give a short and clear explanation of the given image.
- Share a concise interpretation of the image provided.
- Present a compact description of the photo's key features.
- Relay a brief, clear account of the picture shown.
- Render a clear and concise summary of the photo.
- Write a terse but informative summary of the picture.
- Create a compact narrative representing the image presented.
- Please generate a detailed description of the dog in the image, including the breed of the dog, its specific attributes, unique features that can distinguish it from other breeds. (*for ImageNet-Dog*)

## F  ADDITIONAL EXPERIMENTAL RESULTS

In this section, we provide more experimental results that mentioned in the manuscript.

### F.1  MORE POISONED SAMPLE DETECTION RESULTS

- Table 13 shows the detection results on CIFAR-10 with poisoning ratio $\eta = 0.009$, *i.e.*, 50 poisoned samples per class.
- Table 14 shows the detection results on ImageNet-100 with poisoning ratio $\eta = 0.0099$, *i.e.*, 5 poisoned samples per class.

The results show the consistent effectiveness of VDC across different datasets and poisoning ratios.

### F.2  RESULTS OF TRAINING ON THE PURIFIED DATASETS.

- Table 15 shows the normally training results on the purified CIFAR-10 with poisoning ratio $\eta = 0.009$, *i.e.*, 50 poisoned samples per class.
- Table 17 shows the normally training results on the purified CIFAR-10 with poisoning ratio $\eta = 0.09$ noisy ratio $\eta_2 = 0.1$.

The results show that our proposed VDC can indeed improve the reliability and usability of DNNs trained with dirty samples.

Table 11: Examples of label-specific questions on ImageNet-100.

| Label: | **cock** |
|---|---|
| Questions: | Is the object in the image belong to the type of cock? |
| | Is the object in the image a type of poultry? |
| | Is the object in the image commonly found on farms or in rural areas? |
| | Does the object in the image have a comb on top of its head? |
| | Is the object in the image known for its distinctive crowing sound? |
| | Does the object in the image have sharp spurs on its legs? |

| Label: | **goldfinch** |
|---|---|
| Questions: | Is there a type of bird in the image? |
| | Is the object in the image a type of finch? |
| | Does the image feature a small bird known for its vibrant yellow and black coloration? |
| | Is the object in the image a type of finch with bright plumage? |
| | Is the bird in the image a small passerine species? |
| | Is the bird in the image belong to carduelis? |

| Label: | **scorpion** |
|---|---|
| Questions: | Does the object in the image have a venomous stinger? |
| | Is there a scorpion in the image? |
| | Is the object in the image a type of arachnid? |
| | Is the creature in the image venomous? |
| | Is the creature in the image commonly found in desert regions? |
| | Does the image show a scorpion? |

| Label: | **koala** |
|---|---|
| Questions: | Is there a koala in the image? |
| | Does the object in the image have a round face with large, fluffy ears? |
| | Is the object in the image known for its ability to climb trees? |
| | Does the animal in the image primarily feed on eucalyptus leaves? |
| | Is the object in the image typically gray in color? |
| | Is the object in the image commonly found in Australia? |

| Label: | **flamingo** |
|---|---|
| Questions: | Is the image showing a flamingo? |
| | Is the object in the image commonly found in wetland habitats? |
| | Does the object in the image have a long, curved neck? |
| | Is the bird in the image tall with long legs? |
| | Is the bird in the image known for its vibrant pink plumage |
| | Is the bird in the image known for standing on one leg? |

| Label: | **gorilla** |
|---|---|
| Questions: | Is the creature in the image a type of gorilla? |
| | Is the animal in the image known for its intelligence? |
| | Is the animal in the image known for its strength? |
| | Is the animal in the image a primate? |
| | Is the object in the image commonly found in forests or jungles? |
| | Does the object in the image have a large and robust body? |

| Label: | **dumbbell** |
|---|---|
| Questions: | Does the image show a dumbbell? |
| | Is there a dumbbell in the image? |
| | Is the object in the image often used to build muscle strength? |
| | Is the object in the image associated with fitness training? |
| | Is the object in the image commonly hold with hands? |
| | Is the object in the image a type of exercise equipment? |

Table 12: Examples of label-specific questions on ImageNet-100.

| Label: | **hatchet** |
|---|---|
| Questions: | Is there a hatchet in the image? |
| | Does the image show a hatchet? |
| | Is the object in the image a type of cutting tool? |
| | Is the object in the image typically used for chopping? |
| | Is the object in the image often used for splitting wood? |
| | Is the object in the image typically held with one hand? |

| Label: | **stethoscope** |
|---|---|
| Questions: | Is the object in the image a stethoscope? |
| | Does the object in the image have a distinct Y-shaped design? |
| | Is the medical instrument in the image a stethoscope? |
| | Is the device primarily used by doctors? |
| | Is the tool in the image commonly used to listen to heartbeats? |
| | Is the device in the image associated with medical examinations? |

| Label: | **broccoli** |
|---|---|
| Questions: | Is there broccoli in the image? |
| | Is the object in the image a vegetable with a thick, edible stalk? |
| | Is the vegetable in the image broccoli? |
| | Is the vegetable in the image green? |
| | Is the object in the image commonly used in salads? |
| | Is the object in the image often cooked or consumed for its health benefits? |

| Label: | **space shuttle** |
|---|---|
| Questions: | Does the image show a space shuttle? |
| | Is the object in the image capable of launching vertically into space? |
| | Is the object in the image equipped with powerful rocket engines for propulsion? |
| | Is this a spacecraft used to transport astronauts and cargo? |
| | Is the object known for its missions to the International Space Station? |
| | Is this a vehicle that was used by NASA for space exploration? |

| Label: | **pomegranate** |
|---|---|
| Questions: | Is the fruit in the image a pomegranate? |
| | Is the fruit in the image typically red or reddish in color? |
| | Does the fruit in the image have a tough outer rind? |
| | Is the fruit in the image typically used to make juices and other beverages? |
| | Does the object in the image have a crown-like structure at the top? |
| | Does the object in the image have a segmented interior filled with clusters of juicy, ruby-red seeds? |

| Label: | **radio telescope** |
|---|---|
| Questions: | Is the object in the image a type of radio telescope? |
| | Does the object in the image have a large dish or antenna-like structure? |
| | Does the object in the image have a parabolic or spherical reflector to focus radio waves? |
| | Is the device in the image used for radio astronomy? |
| | Is the equipment in the image designed to receive radio waves from the universe? |
| | Does the image depict a device that contributes to radio astronomy research? |

Table 13: Comparison of TPR (%) and FPR (%) for poisoned sample detection on CIFAR-10. $\eta = 0.009$, *i.e.*, 50 poisoned samples per class. Average is the mean of results of different triggers.

| Method | Clean Data | BadNets | | Blended | | SIG | | TrojanNN | | SSBA | | WaNet | | Average | |
|---|---|---|---|---|---|---|---|---|---|---|---|---|---|---|---|
| | | TPR↑ | FPR↓ | TPR↑ | FPR↓ | TPR↑ | FPR↓ | TPR↑ | FPR↓ | TPR↑ | FPR↓ | TPR↑ | FPR↓ | TPR↑ | FPR↓ |
| STRIP | 4% | 86.22 | 11.67 | 4.22 | 10.86 | 99.56 | 11.27 | 99.78 | 9.84 | 65.11 | 9.87 | 2.44 | 9.98 | 59.56 | 10.58 |
| SS | 4% | 97.56 | 12.72 | 99.78 | 12.70 | 100.00 | 12.69 | 3.33 | 13.57 | 99.33 | 12.70 | 92.00 | 12.77 | 82.00 | 12.86 |
| SCAn | 4% | 92.22 | 2.28 | 87.78 | 1.92 | 99.78 | 2.83 | 99.78 | 2.81 | 88.00 | 2.28 | 34.54 | 2.65 | 83.68 | 2.46 |
| Frequency | 4% | 89.11 | 21.51 | 84.22 | 21.55 | 48.67 | 21.76 | 100.00 | 19.32 | 85.56 | 21.66 | 39.78 | 21.72 | 74.56 | 21.25 |
| CT | 4% | 97.56 | 1.32 | 99.50 | 1.66 | 100.00 | 1.01 | 100.00 | 3.92 | 100.00 | 1.82 | 76.00 | 2.58 | 95.51 | 2.05 |
| D-BR | 0% | 0.44 | 0.91 | 0.00 | 0.90 | 0.00 | 0.90 | 11.11 | 0.78 | 1.11 | 0.91 | 1.33 | 0.89 | 2.33 | 0.88 |
| SPECTRE | 0% | 98.00 | 5.91 | 99.78 | 5.90 | 100.00 | 5.89 | 100.00 | 5.89 | 99.33 | 5.90 | 91.56 | 5.97 | 98.11 | 5.91 |
| VDC (Ours) | 0% | 100.00 | 2.72 | 99.56 | 2.72 | 99.78 | 2.72 | 100.00 | 2.72 | 99.78 | 2.72 | 100.00 | 2.72 | 99.85 | 2.72 |

*Dataset: CIFAR-10  $\eta = 0.009$  (50 poisoned samples per class)*

Table 14: Comparison of TPR (%) and FPR (%) for poisoned sample detection on ImageNet-100. $\eta = 0.0099$, *i.e.*, 5 poisoned samples per class. Average is the mean of results of different triggers.

| Method | Clean Data | BadNets | | Blended | | SIG | | TrojanNN | | SSBA | | WaNet | | Average | |
|---|---|---|---|---|---|---|---|---|---|---|---|---|---|---|---|
| | | TPR↑ | FPR↓ | TPR↑ | FPR↓ | TPR↑ | FPR↓ | TPR↑ | FPR↓ | TPR↑ | FPR↓ | TPR↑ | FPR↓ | TPR↑ | FPR↓ |
| STRIP | 4% | 89.70 | 11.94 | 79.39 | 11.36 | 100.00 | 11.03 | 98.99 | 11.09 | 99.60 | 11.61 | 1.62 | 12.50 | 78.22 | 11.59 |
| SS | 4% | 48.08 | 49.92 | 54.14 | 49.86 | 47.47 | 49.92 | 48.08 | 49.92 | 49.49 | 49.90 | 50.91 | 49.89 | 49.70 | 49.90 |
| SCAn | 4% | 96.16 | 2.49 | 87.47 | 1.95 | 88.89 | 2.83 | 98.99 | 1.81 | 86.46 | 2.12 | 97.37 | 2.91 | 92.56 | 2.35 |
| Frequency | 4% | 1.62 | 1.57 | 1.21 | 1.57 | 1.62 | 1.57 | 94.75 | 1.57 | 3.03 | 1.57 | 0.00 | 1.57 | 17.04 | 1.57 |
| CT | 4% | 96.77 | 0.01 | 80.20 | 0.46 | 0.00 | 0.94 | 100.00 | 0.26 | 90.30 | 1.19 | 0.00 | 0.06 | 61.21 | 0.49 |
| D-BR | 0% | 1.01 | 1.99 | 1.41 | 1.65 | 0.00 | 1.98 | 0.81 | 1.81 | 1.21 | 1.77 | 1.21 | 1.82 | 0.94 | 1.84 |
| SPECTRE | 0% | 60.20 | 49.80 | 74.95 | 49.65 | 92.12 | 49.48 | 61.62 | 49.78 | 71.11 | 49.69 | 63.23 | 49.77 | 70.54 | 49.70 |
| VDC (Ours) | 0% | 99.80 | 1.55 | 100.00 | 1.55 | 99.80 | 1.55 | 100.00 | 1.55 | 100.00 | 1.55 | 99.80 | 1.55 | 99.90 | 1.55 |

*Dataset: ImageNet-100  $\eta = 0.0099$  (5 poisoned samples per class)*

Table 15: Comparison of ASR (%) and ACC (%) for training on the purified CIFAR-10 with poisoning ratio $\eta = 0.009$.

| Method | BadNets | | Blended | | SIG | | TrojanNN | | SSBA | | WaNet | | Average | |
|---|---|---|---|---|---|---|---|---|---|---|---|---|---|---|
| | ASR↓ | ACC↑ | ASR↓ | ACC↑ | ASR↓ | ACC↑ | ASR↓ | ACC↑ | ASR↓ | ACC↑ | ASR↓ | ACC↑ | ASR↓ | ACC↑ |
| No detection | 91.83 | 93.67 | 74.00 | 93.63 | 99.64 | 93.50 | 99.99 | 93.56 | 72.86 | 93.70 | 13.18 | 93.37 | 75.25 | 93.57 |
| Strip | 0.73 | 93.27 | 69.97 | 93.19 | 0.27 | 93.15 | 2.7 | 92.02 | 4.53 | 93.7 | 10.79 | 93.03 | 14.83 | 93.06 |
| SS | 0.97 | 92.62 | 0.98 | 92.9 | 0.41 | 92.77 | 99.94 | 92.74 | 1.21 | 92.76 | 0.89 | 93.15 | 17.40 | 92.82 |
| SCAn | 0.6 | 93.38 | 1.59 | 93.09 | 0.23 | 93.19 | 3.92 | 92.89 | 1.52 | 93.62 | 21.44 | 93.73 | 4.88 | 93.32 |
| Frequency | 0.86 | 92.54 | 5.83 | 93.15 | 98.44 | 92.4 | 2.17 | 92.69 | 2.33 | 93.01 | 6.32 | 91.62 | 19.33 | 92.57 |
| CT | 0.79 | 93.24 | 0.71 | 93.94 | 0.12 | 93.7 | 3.96 | 93.17 | 0.57 | 93.76 | 1.32 | 93.55 | 1.25 | 93.56 |
| D-BR | 90.97 | 93.4 | 73.98 | 93.62 | 99.58 | 94.21 | 99.98 | 93.86 | 68 | 93.06 | 18.82 | 93.73 | 75.22 | 93.65 |
| SPECTRE | 0.87 | 92.89 | 1.26 | 92.94 | 0.21 | 92.99 | 4.1 | 92.96 | 1.06 | 92.92 | 1.07 | 92.9 | 1.43 | 92.93 |
| VDC (Ours) | 0.61 | 93.29 | 0.69 | 93.73 | 0.31 | 93.14 | 3.10 | 93.47 | 1.02 | 93.72 | 0.76 | 93.74 | 1.08 | 93.52 |

Table 16: Comparison of ACC (%) for training on the purified datasets with noisy labels.

| Method | **CIFAR-10** $\eta = 0.4$ | | **ImageNe-100** $\eta = 0.4$ | | **ImageNet-Dog** $\eta = 0.4$ | |
|---|---|---|---|---|---|---|
| | Symmetric | Asymmetric | Symmetric | Asymmetric | Symmetric | Asymmetric |
| No detection | 61.84 | 56.09 | 31.21 | 32.65 | 28.45 | 31.35 |
| BHN | 88.71 | 89.21 | 40.12 | 44.25 | 31.65 | 38.90 |
| CORES | 84.68 | 84.70 | 38.41 | 41.87 | 18.70 | 37.05 |
| CL | 87.82 | 57.94 | 39.19 | 46.98 | 18.25 | 30.00 |
| SimiFeat-V | 89.39 | 74.70 | 37.81 | 41.68 | 33.70 | 34.90 |
| SimiFeat-R | 90.48 | 80.54 | 38.31 | 39.70 | 31.86 | 28.80 |
| VDC (Ours) | 90.75 | 90.89 | 66.84 | 69.32 | 46.54 | 48.80 |

Table 17: Comparison of ASR (%) and ACC (%) for training on the purified CIFAR-10 with poisoning ratio $\eta_1 = 0.09$, noisy ratio $\eta_1 = 0.1$.

| | Dataset: CIFAR-10 | | poisoning ratio $\eta_1 = 0.09$ | | noisy ratio $\eta_2 = 0.1$ | | | | | | | | | |
|---|---|---|---|---|---|---|---|---|---|---|---|---|---|---|
| Method | BadNets | | Blended | | SIG | | TrojanNN | | SSBA | | WaNet | | Average | |
| | ASR↓ | ACC↑ | ASR↓ | ACC↑ | ASR↓ | ACC↑ | ASR↓ | ACC↑ | ASR↓ | ACC↑ | ASR↓ | ACC↑ | ASR↓ | ACC↑ |
| No detection | 96.17 | 85.18 | 97.37 | 86.54 | 99.95 | 86.45 | 100.00 | 86.70 | 96.53 | 85.90 | 94.78 | 86.35 | 97.47 | 86.19 |
| Strip | 1.64 | 85.84 | 96.13 | 85.24 | 0.98 | 85.97 | 1.82 | 84.35 | 57.91 | 84.66 | 93.89 | 85.25 | 42.06 | 85.22 |
| SS | 94.38 | 78.33 | 95.31 | 81.33 | 99.96 | 81.58 | 99.97 | 80.56 | 92.81 | 78.45 | 69.07 | 77.34 | 91.92 | 79.60 |
| SCAn | 2.18 | 86.9 | 4.87 | 85.75 | 4.91 | 85.17 | 4.2 | 86.99 | 3.97 | 86.12 | 5.44 | 83.41 | 4.26 | 85.72 |
| Frequency | 75.73 | 85.04 | 76.38 | 83.84 | 99.77 | 84.85 | 3.01 | 85.4 | 72.79 | 82.05 | 89.23 | 84.08 | 69.49 | 84.21 |
| CT | 2.46 | 85.41 | 1.53 | 86.49 | 0.97 | 85.26 | 55.96 | 86.1 | 9.18 | 84.49 | 5.39 | 86.46 | 12.58 | 85.70 |
| D-BR | 90.72 | 86.09 | 96.3 | 86.59 | 99.86 | 86.04 | 100 | 85.46 | 96.59 | 86.16 | 94.93 | 85.11 | 96.40 | 85.91 |
| SPECTRE | 96.71 | 82.51 | 96.89 | 84.62 | 99.91 | 80.34 | 100 | 84.46 | 97.29 | 83.69 | 10.02 | 84.76 | 83.47 | 83.40 |
| BHN | 73.04 | 91.22 | 50.73 | 91.68 | 99.98 | 85.61 | 100.00 | 85.13 | 95.77 | 84.54 | 88.11 | 83.40 | 84.61 | 86.93 |
| CL | 95.86 | 88.92 | 98.10 | 90.29 | 99.97 | 86.01 | 99.99 | 85.58 | 96.38 | 85.12 | 91.19 | 84.19 | 96.92 | 86.69 |
| CORES | 95.88 | 81.94 | 96.53 | 84.98 | 99.96 | 85.40 | 100.00 | 86.11 | 95.18 | 84.42 | 93.90 | 84.22 | 96.91 | 84.51 |
| SimiFeat-V | 1.21 | 92.38 | 71.01 | 92.21 | 99.96 | 84.76 | 99.99 | 85.42 | 95.74 | 84.07 | 92.97 | 84.35 | 76.81 | 87.20 |
| SimiFeat-R | 1.04 | 92.79 | 67.91 | 92.34 | 99.97 | 85.23 | 99.99 | 85.05 | 95.46 | 84.47 | 88.73 | 82.90 | 75.52 | 87.13 |
| VDC (Ours) | 1.01 | 92.58 | 1.13 | 91.73 | 3.07 | 91.67 | 4.59 | 92.79 | 1.39 | 92.06 | 0.99 | 92.63 | 2.03 | 92.24 |

