# OpenReview forum: "VDC: Versatile Data Cleanser based on Visual-Linguistic Inconsistency by Multimodal Large Language Models"
_ICLR.cc/2024/Conference — ICLR 2024 poster_

### Official Review · Reviewer_vuG8 · 2023-10-29

**Soundness:** 2 fair
**Presentation:** 3 good
**Contribution:** 2 fair
**Rating:** 6
**Confidence:** 5

**Summary:**

The paper proposes a general "label" cleaning/filtering approach and aims to detect three types of errors: "poisoned samples", "noisy samples", and "hybrid dirty samples". The paper takes advantage of the exceptional capability of multimodal large language model (MLLM) and casts the error detection problem into a three-step scoring pipeline. The pipeline consists of 1) visual question generation, 2) visual question answering, and 3) visual answer evaluation. The main argument is that, unlike prior arts, the proposed approach can detect all three types of label errors and achieves better performance in common benchmarks, including ImageNet-100 and CIFAR-10.

**Strengths:**

- The proposed method leverages the recent trend of MLLM to the label error detection literature
- The proposed method is training-free

**Weaknesses:**

- The empirical comparison is not fair.
    - The propose approach is using instruct-BLIP (larger network trained on larger dataset), while the baseline is usually using less-expressive network trained on smaller datasets, e.g., CL is using ResNet-18 and trained on CIFAR dataset.
- The claim that the proposed method mitigates all three types of label noises is too strong.
    - The proposed approach is general, but so do other baselines. For example, SimiFeat-V leverages the feature similarity to detect noisy labels, which is also applicable to the scenario of “poisoned samples” as long as the feature extractor is trained on a poison-free dataset.

**Questions:**

1. The name of “visual question generation” is confusing. From my understanding to the paper, there is no visual signal in this stage. Can the author confirm this?
2. What are the accuracy of each question?
    - Figure 2 shows the TRR of the general and label-specific questions. I am curious of what specific question is challenging for the instruct-BLIP. Is there any specific question that is always challenging to the instructBLIP and, therefore, removing those questions actually help the overall performance
3. Given that the method leverages a ensemble of the MLLM responses, instead of taking the average accuracy as score as in Eq. 4, does the author think that using techniques like label aggregation further improves the overall performances?
4. To make a fair comparison with other baselines, I suggest the authors could compare baselines with MLLM as well. For example, CL usually trains their classifier with leave-one-out or cross validation, which limits the size of the train dataset. However, one can also uses off-the-shelf classifier and apply CL on it.
5. Do the authors consider the data-leakage problem? Does the train dataset used to train the instruct-BLIP accidentally include the images in ImageNet or CIFAR? If so, how do accurately validate the performances?
6. Do the authors observe any accumulated error in the three-step pipeline? For example, the label-specific question include some noisy questions, which leads to poisoned answers in the second stage.

---

> ### Author Response · Authors · 2023-11-17
> **Response to Reviewer vuG8 (Part 1/3)**
>
> Dear Reviewer vuG8,
>
> We sincerely appreciate your precious time and constructive comments, and we are greatly encouraged by your high recognition of **the innovation of our approach, the unified treatment of noisy data and poisoned data, and extensive experiments**.
>
> In the following, we would like to answer your concerns separately.
>
> ---
> **Q1:** The empirical comparison is not fair. The propose approach is using instruct-BLIP (larger network trained on larger dataset), while the baseline is usually using less-expressive network trained on smaller datasets, e.g., CL is using ResNet-18 and trained on CIFAR dataset.
>
> **R1:** Thanks for your constructive feedback. We would like to clarify this concern from the following points:
>
> - **Determining the fairness of the comparison hinges on whether impediments are imposed on our method. We avoid employing resources inaccessible to others.** Currently, the barriers for utilizing large models have significantly diminished, granting universal access to the capabilities of large  models through APIs. We incorporated this innovative tool into the realm of this challenging task.
> - **We discover that the fundamental commonality of dirty samples lies in visual-linguistic inconsistency.** Grounded in this notion, we propose a pioneering framework adept at harnessing the full potential of visual and language understanding capabilities of large models. **Previous methods neglected this commonality,** required training on noisy datasets, and struggled to harness the potential of large models effectively. This represents a distinctive advantage of  our method. It is reasonable to compare the advantages of our method with previous methods.
>
> ---
>
> **Q2:** The claim that the proposed method mitigates all three types of label noises is too strong.
> The proposed approach is general, but so do other baselines. For example, SimiFeat-V leverages the feature similarity to detect noisy labels, which is also applicable to the scenario of “poisoned samples” as long as the feature extractor is trained on a poison-free dataset.
>
> **R2:** Thanks for your constructive comment.  We would like to clarify from the following aspects:
>
> - **To the best of our knowledge, we are the first work that designed for detecting three types of dirty samples simultaneously  and empirically evaluated in the literature**. Previous methods were primarily designed for either noisy labels or poisoned samples, lacking consideration for the coexistence of hybrid dirty samples within the dataset. Moreover, **none of them capture the fundamental commonality of such dirty samples -- visual-linguistic inconsistency.**
> - **In our implementation, follow the original paper of SimiFeat, we indeed adopt the image encoder of CLIP as the feature extractor, which is a poison-free method.** Experimental results (see Table 5 in the manuscript) show that SimiFeat  still has a large performance gap of about 25% compared to our LPS, which mainly due to the representations of poisoned samples and benign samples are quite different.
>
>
> ---
>
> **Q3:** The name of “visual question generation” is confusing. From my understanding to the paper, there is no visual signal in this stage. Can the author confirm this?
>
> **R3:** Thanks for this  valuable suggestion. We name this module as "Visual Question Generation" because these questions are regarded as "visual questions" for the second  "Visual Question Answering (VQA)" module. We note that this name may confuse readers, so we will rename this module as "Question Generation" in the revised version.
>
> ---

---

> ### Author Response · Authors · 2023-11-17
> **Response to Reviewer vuG8 (Part 2/3)**
>
> ---
>
> **Q4:** Figure 2 shows the TPR of the general and label-specific questions. I am curious of what specific question is challenging for the instruct-BLIP. Is there any specific question that is always challenging to the instructBLIP and, therefore, removing those questions actually help the overall performance.
>
> **R4:** Thanks for this constructive comment. Follow your suggstion, we conduct additional experiment to analyze the importance of different questions.
>
> - **Experiment settings:** We choose 1 target label and 3 clean label from ImageNet-100. For target label, we calculate the the accuracy of each question (answer with false) from all poisoned samples. For other clean labels, we calculate the the accuracy of each question (answer with true) from benign samples from the corresponding labels.
> - **Analysis:**
>     - **We find that there are indeed some questions challenging for InstructBLIP for these labels.**  For example, the geographic questions requiring geography knowledge are relatively more challenge.
>     - **We have considered the existing of challenging questions and adopt vote-based ensemble to avoid negative effect.** We remove the last questions of target label, the TPR changes from 99.93% to 99.91%. We also remove the last question of wolf spider, and the FPR of this class changes from 1.8% to 1.4%. The results indicate that the existing of few challenging questions does not have a severe impact on performance.
>
>
> **Target Label: cock**
> | Question| Acc of poisoned sampels  |
> | - | - |
> | Is the object in the image belong to a type of cock?| 0.988  |
> | Does the image feature a cock?| 0.971  |
> | Does the bird in the image typically have a distinctive comb on its head?| 0.942  |
> |Is the object in the image commonly raised for cockfighting in some cultures?|0.91
> | Is the object in the image associated with the dawn and often known for crowing at sunrise?| 0.844  |
> | **Is the object in the image a domesticated fowl commonly found on farms?** | 0.82 |
>
>
> **Clean Label: tree frog**
>
> | Question| Acc of benign sampls  |
> | - | - |
> | Is there a type of frog shown in the image?| 0.95  |
> | Is the creature in the image a type of frog?| 0.95  |
> | Does the image feature a frog?| 0.95  |
> | Is the object in the image belong to amphibian?| 0.90  |
> | Is the creature in the image capable of croaking?| 0.90  |
> | **Is the animal of the image commonly found near the water?** | 0.645 |
>
>
> **Clean Label: centipede**
>
> | Question| Acc of benign sampls   |
> | - | ----- |
> | Is the creature in the image a type of arthropod?| 0.992 |
> | Is the creature in the image known for its many legs?| 0.95  |
> | Is the object in the image a type of centipede?| 0.972 |
> | Does the image feature a centipede?| 0.972 |
> | Is there a centipede in the image?| 0.90  |
> | **Is the object in the image typically found in damp environments?** | 0.792 |
>
> **Clean Label: king penguin**
>
> | Question| Acc of benign sampls   |
> | - | - |
> | Is the bird in the image known for its black and white plumage? | 0.95  |
> | Can the animal in the image survive in  cold temperatures?   | 0.935 |
> | Is there a penguin in the image?| 0.93  |
> | Does the image feature a penguin?| 0.93  |
> | Is the object in the image a type of penguin?| 0.93  |
> | **Is the animal in the image commonly found in Antarctica?** | 0.848 |
>
> ---

---

> ### Author Response · Authors · 2023-11-17
> **Response to Reviewer vuG8 (Part 3/3)**
>
> ---
>
> **Q5:** Given that the method leverages a ensemble of the MLLM responses, instead of taking the average accuracy as score as in Eq. 4, does the author think that using techniques like label aggregation further improves the overall performances?
>
> **R5:** Thanks for this insightful comment.
>
> - **Actually, in our paper, we adopt majority vote in the vote-based ensemble instead of taking the average accuracy** (see Eq(4) in the manuscript). We find that this baseline ensemble method is sufficient for our framework.
> - **Taking other label aggregations is an insightful suggestion.** In our further work, we will consider the importance of different questions and taking weight-based label aggragation to further improve the overall performance.
>
> ---
>
> **Q6:** To make a fair comparison with other baselines, I suggest the authors could compare baselines with MLLM as well. For example, CL usually trains their classifier with leave-one-out or cross validation, which limits the size of the train dataset. However, one can also uses off-the-shelf classifier and apply CL on it.
>
> **R6:** Thanks for your constructive comments. Following your suggestion, we replace the backbone classifier of CL as pre-trained CLIP and the results on CIFAR-10 are shown in the following table. We can see that even though using off-the-shelf classifier, **our proposed method still outperforms CL+CLIP by a considerable gap**, mainly due to the inevitable error in the predicted probability of CLIP.
>
> ||Symmetric (TPR %)| Symmetric (FPR %)|Asymmetric (TPR %)|Asymmetric (FPR %)|
> |-| -|-|-|-|
> | CL| 85.05| 8.75| 82.49| 4.50|
> | CL+CLIP|  89.38|  5.80| 89.89| 5.89|
> | **VDC (Ours)** | 98.81| 2.61| 99.60| 2.62|
>
> ---
>
> **Q7:** Do the authors consider the data-leakage problem? Does the train dataset used to train the instruct-BLIP accidentally include the images in ImageNet or CIFAR? If so, how do accurately validate the performances?
>
> **R7:** Thank your this insightful comment.
> - **The training datasets used to train the InstructBLIP do not include ImageNet and CIFAR.** We check the publised paper of InstrcutBLIP (see Figure 2) and find that the collected datasets in InstructBLIP used for vision-language instruction tuning exclude ImageNet and CIFAR.
> - **A native validation method is directly using InstrcutBLIP to perform classification task on CIFAR-10.** The used prompt is '*What’s the category of the image? Please choose one category from the following 10 labels ["airplane","automobile","bird","cat","deer","dog","frog","horse","ship","truck"]*'. The classification accuracy is only **69.36%**. **It turns out that directly applying large models is not sufficient to solve downstream tasks and requires additional non-trivial careful design. This further emphasizes the unique contribution of our work.**
>
> ---
>
> **Q8:** Do the authors observe any accumulated error in the three-step pipeline? For example, the label-specific question include some noisy questions, which leads to poisoned answers in the second stage.
>
> **R8:** Thank you for this suggetion. We would like to clarify from the following aspects:
> - **We have considered this accumulated error in our proposed framework.** Actually, the voted-based ensemble in the visual answer evaluation module is designed for preventing this phenomenon.
> - **Our proposed method is robust for noisy questions.** To evaluate it, we artificially replace the last questions of all poisoned samples as wrong answers on CIFAR-10 with BadNets, and the TPR slightly changes from 99.93% to 99.56%, demonstrating that our proposed method is robust for noisy questions.
>
> ---
>
> Thanks again for your time and attention. We hope the response can address your concerns.
>
> Best regards,
>
> Authors

---

> > ### Comment · Reviewer_vuG8 · 2023-11-21
> > **Response**
> >
> > Appreciate the authors' further clarification.
> >
> > 1. **The use of foundation models**: I am not complaining the use of foundation models (instruct-BLIP). Instead, I want to point out that the improvements not only come from the approach, but also from the strong visual backbones. The authors should either make this point clear in the submission or adopt the same visual backbone for other baseline approaches.
> > 2. **Experiments on the question types**: The analysis is great and makes a lot of senses.
> > 3. **Experiments on CL+CLIP**: Appreciate for the additional efforts from the authors. The improvements are more convincing to me now.
> >
> > Overall, the authors did a great job clarifying my questions and the additional experiments are convincing that the proposed approach does improve in this setting. Please include these analysis in the paper. I am raising my score to 6.

---

> > > ### Author Response · Authors · 2023-11-21
> > > **Appreciation for Your Feedback**
> > >
> > > Dear Reviewer vuG8:
> > >
> > >
> > > Thank you for your positive feedback on our paper.
> > >
> > > Engaging in a constructive dialogue with you has been both meaningful and enjoyable, which allowed me to better explain the strength and theory of our proposed method.
> > >
> > >
> > > Your valuable suggestions are highly regarded, and we are committed to enhancing the manuscript by incorporating additional analyses and experiments. Your input is instrumental in refining the quality of our work, and we are grateful for your thoughtful and thorough review.
> > >
> > > Thank you once again for your time and effort in evaluating our paper.
> > >
> > >
> > > Best regards,
> > >
> > > Authors

---

### Official Review · Reviewer_481L · 2023-10-31

**Soundness:** 3 good
**Presentation:** 3 good
**Contribution:** 2 fair
**Rating:** 6
**Confidence:** 4

**Summary:**

To address common noise data and backdoor attack problems in deep learning, the paper designed a data cleaning tool called VDC using a multimodal large language model. By designing generalized visual question answering questions and label-specific visual question answering questions, VDC can remove dirty and poisoned data from the dataset based on the inconsistency between the semantic of the obtained question results and the semantic of the image itself. The paper compared VDC with backdoor attack methods and noisy learning methods, demonstrating the effectiveness of VDC.

**Strengths:**

+ The paper's approach is innovative, using popular multimodal large models to replace manual data cleaning work.
+ The paper unified the treatment of noisy data and poisoned data from backdoor attacks, which is relatively rare in previous research.
+ The paper's extensive experiments demonstrate the effectiveness of VDC in handling dirty data.

**Weaknesses:**

+ The overall content of the paper seems to be more about using out-of-distribution methods to filter data, and perhaps this should be reflected in the accuracy of sample selection.
+ While the paper effectively addresses the issue of dirty data using MLLM, it seems to have a bias towards reporting the application of MLLM.

**Questions:**

+ How is the threshold for excluding dirty data determined in the paper?
+ Has the author considered the scenario where MLLM itself serves as a detector rather than a data cleaner?
+ The paper's method relies heavily on MLLM, which in essence depends on more data and better labeling. Therefore, it may not be necessary to test models that use more clean data on small datasets, as this may go against the goal of advancing deep learning.

---

> ### Author Response · Authors · 2023-11-17
> **Response to Reviewer 481L (Part 1/2)**
>
> Dear Reviewer 481L,
>
> We sincerely appreciate your precious time and constructive comments, and we are greatly encouraged by your high recognition of the innovation of our approach, the unified treatment of noisy data and poisoned data, and extensive experiments.
>
> In the following, we would like to answer your concerns separately.
>
> ---
>
> **Q1:** The overall content of the paper seems to be more about using out-of-distribution methods to filter data, and perhaps this should be reflected in the accuracy of sample selection.
>
> **R1:** Thank you for your constructive comments. Although the multimodal large language model (MLLM) may be out of distribution compared to the noisy dataset at hand, **its remarkable generalization capabilities make it superior to most previous methods in the task of detecting dirty samples**, as demonstrated by the experimental results in the manuscript, **showcasing the effectiveness of our approach.**
>
> ---
>
> **Q2:** While the paper effectively addresses the issue of dirty data using MLLM, it seems to have a bias towards reporting the application of MLLM.
>
> **R2:** Thanks for this insightful comment. We would like to explain from the following aspects.
>
> - **This task is crucial and challenging in real-life scenarios.** As Reviewer N7YG pointed out, "Noisy or dirty data detection and cleaning is an **important** research topic. It is becoming even more **critical** for recent machine learning research.". Furthermore, the presence of hybrid dirty samples in real-world scenarios poses practical and challenging aspects that surpass the capabilities of established methods. Our paper is dedicated to addressing this challenging task rather than merely extending the application of large models.
> - **We present a new perspective for detecting dirty samples in this field.** We find a notable commonality of noisy labels and poisoned samples lies in visual-linguistic inconsistency, which was overlooked by previous methods. **This revelation can inspire more researchers to continue exploring follow this direction, marking the most significant contribution of our paper to this field.**
> - **Based on the commonality that we discovered, we propose a novel  framework for addressing the task.** We build a bridge between the visual and language foundation models and this task. We observe that the multimodal large language models have excellent visual and language understanding capabilities, which are suitable for the visual-linguistic inconsistency that we discovered. Hence, we propose a novel framework to fully mine this specific potential of large models.
> - **The proposed framework is simple, user-friendly, training-free and effective, and not just the  application of large models.** Based on the commonality of dirty samples and the characteristics of large models, we novelly designe this framework to fully utilize the abilities of large models to solve this task.
>
>
> Consequently, the primary contribution of our work lies not in the mere application of large model but in the proposal of visual-linguistic inconsistency as a key factor and the strategic incorporation of large models to address this task.
>
> ---

---

> ### Author Response · Authors · 2023-11-20
> **Response to Reviewer 481L (Part 2/2)**
>
> ---
>
> **Q3:** How is the threshold for excluding dirty data determined in the paper?
>
> **R3:** Thank you for this question. As detailed in Section 5.1 of the manuscript, the threshold $\alpha$ is set as 0.5 across all experiments.
>
> ---
>
> **Q4:** Has the author considered the scenario where MLLM itself serves as a detector rather than a data cleaner?
>
> **R4:** Thank you for your insightful comment.  I find some ambiguity in the term "detector" as used in your response. Our proposed VDC serves the purpose of identifying dirty samples without necessitating the training of a classifier on the primary dataset. **Consequently, the VDC functions as a zero-shot detector.**
>
> ---
>
> **Q5:** The paper's method relies heavily on MLLM, which in essence depends on more data and better labeling. Therefore, it may not be necessary to test models that use more clean data on small datasets, as this may go against the goal of advancing deep learning.
>
> **R5:** Thanks for your constructive comment. We would like to explain from the following aspects:
>
> - **This paper focuses on solving this challenging task in the real world.** Recognizing the inherent commonalities present in dirty samples serves as the foundation of our approach. Based on this valuable revelation, we then propose a novel framework that leverages large-scale models as an effective tool to solve this task.
> - **Integrating large models to solve domain-specific downstream tasks is expected to become a mainstream trend.** Significantly, advancements in large model technology have facilitated widespread access to their capabilities for ordinary users at relatively economical costs. Our exploration extensively delves into the potential of large models in addressing visual-linguistic inconsistency, thereby fostering progress in the realms of backdoor learning and robust learning.
>
> ---
>
> Thanks again for your constructive comments and your recognition of our efforts. We hope the response can address your concerns.
>
> Best regards,
>
> Authors

---

> ### Author Response · Authors · 2023-11-21
> **Appreciation for Review and Request for Feedback**
>
> Dear Reviewer 481L,
>
> We want to convey our sincere appreciation for the valuable insights and suggestions you provided regarding our work.
>
> We have made efforts to address the concerns and queries you raised during the rebuttal process. It would be immensely helpful to receive feedback on whether our response effectively alleviated any doubts you may have had. Your feedback is crucial to enhancing the quality of our work.
>
> Recognizing the demands of your busy schedule, we genuinely appreciate your contribution to the refinement of our manuscript. As the end of the rebuttal period is approaching, we eagerly await your reply before the end.
>
> Once again, thank you for your time and consideration.
>
> Best regards,
>
> Authors

---

> ### Author Response · Authors · 2023-11-22
> **A Kind Reminder of the Final Feedback**
>
> Dear Reviewer 481L,
>
> Thanks for your assessment and your encouragement of our work. As the deadline for reviewer-author discussion is approaching, we would be happy to take this opportunity to make more discussions about any of our questions or concerns. If our response has addressed your concerns, we would be grateful if you could re-evaluate our paper based on our feedback.
>
> Sincerely,
>
> Authors

---

### Official Review · Reviewer_N7YG · 2023-10-31

**Soundness:** 3 good
**Presentation:** 3 good
**Contribution:** 2 fair
**Rating:** 6
**Confidence:** 3

**Summary:**

This paper proposes a working pipeline for noisy data detection. Compared with existing works focusing on noisy data or noisy label, this work aims to obtain an integrated framework to handle a comprehensive scenario including various noisy cases. Specifically, it terms it as visual-language inconsistency. Leveraging on several prompt techniques, the proposed framework achieves promising results compared with other baselines.

**Strengths:**

1. Noisy or dirty data detection and cleaning is an important research topic. It is becoming even more critical for recent machine learning research since the data scale is always getting larger.
2. The proposed framework wisely utilize the advantage of current large-scale model to benefit the dirty data detection task.
3. Comprehensive empirical results show the framework superiority compared with other baselines.
4. The whole draft is in a good format for readers.

**Weaknesses:**

1. I mainly concern about the technical contribution in this draft. The wise combination of prompting and dirty data detection is interesting. However, it still based on the visual-language understanding from large-scale pretrained model. Only based on such powerful tools relatively diminish this paper novelty. In addition, the key point of this paper is proposing an integrated detection pipeline instead of only focusing on sample or label. This point looks like a trivial combination which is incremental compared with previous settings. Is this setting practical and necessary for real-world scenarios?
2. Adding more recent published works for comparison may further help to support the paper contribution. Currently, only one or two baselines are published within past one year.

**Questions:**

Please refer to the strengths and weakness for details. Even if I am concerning some points in weakness, I recognize other aspects of this paper mentioned in strengths. Overall, I lean to vote for an acceptance for now and I would like to check other reviewers' comments to discuss and make my final decision.

---

> ### Author Response · Authors · 2023-11-17
> **Response to Reviewer N7YG (Part 1/2)**
>
> Dear Reviewer N7YG,
>
> We sincerely appreciate your precious time and constructive comments, and we are greatly encouraged by your high recognition of the following aspects:
> - The **significance** of noisy or dirty data detection and cleaning.
> - The **optimal utilization** of large-scale models.
> - The empirical **superiority** over baselines.
> - The **reader-friendly** format of the draft.
>
> In the following, we would like to answer your concerns separately.
>
> ---
>
> **Q1:** I mainly concern about the technical contribution in this draft. The wise combination of prompting and dirty data detection is **interesting**. However, it still based on the visual-linguistic understanding from large-scale pretrained model. Only based on such powerful tools relatively diminish this paper novelty. In addition, the key point of this paper is proposing an integrated detection pipeline instead of only focusing on sample or label. This point looks like a **trivial** combination which is incremental compared with previous settings. Is this setting practical and necessary for real-world scenarios?
>
> **R1:** Thank you for this constructive comment. We would like to share our thoughts from the following aspects:
> - **Dirty samples is a practical threat in the real-world scenarios.** For poisoned samples, malicious attackers intentionally manipulate partical clean samples to inject backdoor by embedding triggers and changing the ground-truth labels. For noisy labels, human annotators or automatic annotation robots may make mistakes accidentally in the crowdsourcing or web crawling. Therefore, **detecting dirty samples is imperative for real-world scenario.**
> - **We present a new perspective for detecting dirty samples in this field.** We find a notable commonality of noisy labels and poisoned samples lies in visual-linguistic inconsistency, which was overlooked by previous methods. This revelation can inspire more researchers to continue exploring follow this direction, marking the most significant contribution of our paper to this field.
> - **Based on the commonality that we discovered, we propose an easy, user-friendly, training-free and effective framework for addressing the task.**  We build a bridge between the visual and language foundation models and this task.  We observe that the multimodal large language models have excellent visual and language understanding capabilities, which are suitable for the visual-linguistic inconsistency that we discovered. Hence, we propose a novel framework to fully mine this specific potential of large models.
> - **The framework that leverages large-scale models is practical in the real-world scenarios:** Recently, with the increasing availability of LLM and MLLM accessible via APIs, VDC can leverage the power of these large models to accomplish the task at relatively low cost (e.g., GPT-4) for ordinary users. Therefore, our method is training-free and does not require training on the entire noisy dataset like previous methods, eliminating the need for extensive computing resources, making our method very practical for ordinary users in the real-world scenarios.
>
> In summary, while our technical contribution may not be deemed revolutionary, we have introduced a novel perspective and innovative tools to effectively address the task at hand. This substantial contribution merits due consideration and should not be overlooked.
>
> ---

---

> ### Author Response · Authors · 2023-11-20
> **Response to Reviewer N7YG (Part 2/2)**
>
> ---
>
> **Q2:** Adding more recent published works for comparison may further help to support the paper contribution. Currently, only one or two baselines are published within past one year.
>
> **R2:** Thanks for this constructive comment. Following your suggestion, We have found  two  other papers published at CVPR 2023 and ICML 2023 for comparison [1, 2].
>
> - **Introduction of baselines:** FREAK[1] is a frequency-based poisoned sample detection algorithm. LogitClip[2] enhances the noise robustness of existing losses by clamping the norm of the logit vector, which can be combined with CORES to filter noise labels and then improve the accuracy of the trained model, as compared in the draft.
> - **Experimental setting:** For FREAK, we add evaluation on poisoned sample detection on CIFAR-10 with BadNets and Blended attacks, where poisoning ratio $\eta$ is set as 0.09. For LogitClip, we combine it with Cores and evalute on noisy label detection on CIFAR-10. The results are shown in **Table 1 and 2** as follows respectively.
> - **Analysis:**  The results show that VDC outperforms other methods. We can obtain the same conclusion as in the draft: **VDC consistently exhibits superior performance to baselines.**
>
>
> Table 1: TPR and FPR results of poisoned sample detecion on CIFAR-10.
> || BadNets (TPR %) | BadNets (FPR %) | Blended (TPR %) | Blended (FPR %) |
> | - | - | - | - | - |
> | FREAK [1]| 95.70| 2.98| 10.23| 3.46|
> | **VDC (Ours)** | 99.93| 2.75|99.87|2.75|
>
> Table 2: Testing accuracy after filtering noisy label on CIFAR-10.
> || Symmetric Noise (ACC %) | Asymmetric Noise (ACC %) |
> | - | - | - |
> | Cores| 84.68| 84.70|
> | Cores + LogitClip [2] | 85.84| 86.28|
> | **VDC (Ours)**| 90.75| 90.89|
>
>
> Thanks again for your constructive comments and your recognition of our efforts. I hope this response can address your concerns.
>
> Best regards,
>
> Authors
>
> ---
>
> **Refenence**
>
> [1] Don’t FREAK Out: A Frequency-Inspired Approach to Detecting Backdoor Poisoned Samples in DNNs, In CVPR 2023
>
>
> [2] Mitigating Memorization of Noisy Labels by Clipping the Model Prediction, In ICML 2023

---

> ### Author Response · Authors · 2023-11-21
> **Appreciation for Review and Request for Feedback**
>
> Dear Reviewer N7YG,
>
> We want to convey our sincere appreciation for the valuable insights and suggestions you provided regarding our work.
>
> We have made efforts to address the concerns and queries you raised during the rebuttal process. It would be immensely helpful to receive feedback on whether our response effectively alleviated any doubts you may have had. Your feedback is crucial to enhancing the quality of our work.
>
> Recognizing the demands of your busy schedule, we genuinely appreciate your contribution to the refinement of our manuscript. As the end of the rebuttal period is approaching, we eagerly await your reply before the end.
>
> Once again, thank you for your time and consideration.
>
> Best regards,
>
> Authors

---

> ### Author Response · Authors · 2023-11-22
> **A Kind Reminder of the Final Feedback**
>
> Dear Reviewer N7YG,
>
> Thanks for your assessment and your encouragement of our work. As the deadline for reviewer-author discussion approaches, we would be happy to take this opportunity to make more discussions about any of our questions or concerns. If our response has addressed your concerns, we would be grateful if you could re-evaluate our paper based on our feedback.
>
> Sincerely,
>
> Authors

---

### Author Response · Authors · 2023-11-20
**A common response to the concerns about the contribution of this paper**

Dear Reviewers and AC,

Thank you for your time and efforts in handling our submission. We appreciate the professional feedback and suggestions from the reviewers, and we are trying our best to address their concerns in the rebuttal. Here, we would like to provide a common response to the concerns about the contribution of our paper.


- **First of all, the dirty sample detection task solved in our paper is curical, practical, and challenging in the real world.**
    - **The existing of dirty samples makes the DNNs vulnerable and unreliable.** (1) For poisoned samples, DNNs trained on the poisoned dataset will be injected with backdoor, i.e., predict any poisoned sample as the target label during the inference stage. (2) For noisy labels, training DNNs using the noisy dataset will significantly degrade the overall performance.
    - **Dirty samples is a practical threat in the real-world scenario.** (1) For poisoned samples, malicious attackers intentionally manipulate partical clean samples to inject backdoor by embedding triggers and changing the ground-truth labels. (2) For noisy labels, human annotators or automatic annotation robots may make mistakes accidentally in the crowdsourcing or web crawling.
    - **Detecting dirty sample  is still a challenging task that reamins to be solved.** Even though some poisoned sample detectors for backdoor defense and noisy label detectors for robust machine learning have been proposed in the literature, none of them focus on both poisoned samples and noisy labels simultaneously.
- **We present a new perspective for detecting dirty samples in this field.**
    - **We find a notable commonality of noisy labels and poisoned samples lies in visual-linguistic inconsistency**, i.e., the semantics of the visual content of the image is inconsistent with the semantics of the corresponding labels. This fundamental commonality was overlooked by previous methods.
    - **This revelation can inspire more researchers to continue exploring follow this direction, marking the most significant contribution of our paper to this field.**
- **Moreover, based on the commonality that we discovered, we propose an easy, user-friendly, training-free and effective framework for addressing the task.**
    - **We build a bridge between the visual and language foundation models and this task.**   We observe that the multimodal large language models have excellent visual and language understanding capabilities, which are suitable for the visual-linguistic inconsistency that we discovered. Hence, we propose a novel framework to fully mine this specific potential of large models.
    - **Our proposed method is user-friendly and training-free.** With the increasing availability of LLM and MLLM accessible via APIs, our method can leverage the power of large models to accomplish the task at relatively low cost.  Furthermore, VDC does not require training on the entire noisy dataset like previous methods, eliminating the need for extensive computing resources, making our method very practical for oridinary users.
    - **Our proposed method achieves significantly enhancements compared to baselines.** Moreover, with the rapid development of large-scale models, the performance of VDC will be further improved in the future.
- **In the realm of large model development, our investigation emphasizes the efficacy of multimodal large models in addressing critical issues pertaining to data cleansing and AI security.** This not only introduces a new application paradigm for large models but also provides additional momentum for their ongoing advancement. Furthermore, the robust progression of large models inherently stands to gain from  these applications.

I hope this response can help you appreciate the contribution of our paper.

Best regards,

Authors

---

### Meta-Review · Area_Chair_JGV8 · 2023-12-04

**Metareview:**

This paper considers the problem of noisy data detection. To solve this problem, an integrated framework is proposed to handle a comprehensive scenario. The proposed framework leverages several prompt techniques and achieves promising results.

**Strengths**
- The paper studies an important topic in the community.
- It is interesting to utilize the advantage of current large-scale model to benefit the dirty data detection task.
- Comprehensive experiments are provided to demonstrate the effectiveness of the proposed method over previous methods.
- The paper is well-written and easy to follow.

**Weaknesses**

Pre-rebuttal, many concerns are raised, including more fair comparison, clarification of the novelty, more experiments, etc. Most of them have been addressed in the rebuttal. But two concerns still remain.

- Lacks 1) analysis on why the proposed method is better, 2) in what scenario, the proposed method tends to work better, and finally 3) and why MLLM is preferred in this application.

- Make the using of foundations models more clear.

**Justification For Why Not Higher Score:**

Lack of explanations of the following aspects.

- Lacks 1) analysis on why the proposed method is better, 2) in what scenario, the proposed method tends to work better, and finally 3) and why MLLM is preferred in this application.

- Make the using of foundations models more clear.

**Justification For Why Not Lower Score:**

This paper proposes an interesting, effective framework to solve an important setting. Extensive experiments demonstrate the benefits of the proposed method.

---

### Decision · Program_Chairs · 2024-01-16

Accept (poster)